# SENSITIVITY-INFORMED REGULARIZATION FOR OFFLINE BLACK-BOX OPTIMIZATION

## ABSTRACT

Offline optimization is an important task in numerous material engineering domains where online experimentation to collect data is too expensive and needs to be replaced by an in silico maximization of a surrogate of the black-box function. Although such a surrogate can be learned from offline data, its prediction might not be reliable outside the offline data regime, which happens when the surrogate has narrow prediction margin and is (therefore) sensitive to small perturbations of its parameterization. This raises the following questions: (1) how to regulate the sensitivity of a surrogate model; and (2) whether conditioning an offline optimizer with such less sensitive surrogate will lead to better optimization performance. To address these questions, we develop an optimizable sensitivity measurement for the surrogate model, which then inspires a sensitivity-informed regularizer that is applicable to a wide range of offline optimizers. This development is both orthogonal and synergistic to prior research on offline optimization, which is demonstrated in our extensive experiment benchmark.

## 1 INTRODUCTION

Finding material designs that maximize a set of desirable properties is a fundamental task in material engineering. Historically, these design problems were frequently tackled through online experimentation, which can be exceedingly labor-intensive, time-consuming, and often impractical. To avoid such expenses, offline optimization (Brookes et al., 2019; Trabucco et al., 2021; 2022) has emerged as a computational alternative that leverages past experiment results to predict properties of unseen material candidates without running actual experiments. This is achieved via (1) fitting a parameterized model on such past data relating the material input with its output properties; and (2) finding an input optimizer with respect to the learned parameterization.

Naively, such in silico approach would trivialize the optimal design problem into a vanilla application of gradient ascent and supervised learning. However, in practice, the prediction of such vanilla surrogate might not be reliable outside the offline data regime (Fannjiang & Listgarten, 2020). Often, its prediction can become highly erratic at out-of-distribution data regimes, misguiding the optimization process toward sub-optimal candidates. This happens when the surrogate has narrow prediction margin at those out-of-distribution input regimes where a small perturbation to the model weights might cause a significant change in its prediction. This is potentially due to the fact that without relevant data, the training process does not have any mechanisms to recognize and avoid moving toward such sensitive model candidates.

To date, while most existing approaches addressing this problem have proposed numerous strategies to avoid making such spurious predictions during extrapolation (due to high model sensitivity), their conditioning techniques were not built on a direct characterization of sensitivity. For example, Fu & Levine (2021); Kumar & Levine (2020); Trabucco et al. (2021; 2022) and Yu et al. (2021) reduce their surrogate estimations at out-of-distribution inputs, while Brookes et al. (2019) and Fannjiang & Listgarten (2020) condition on domain-specific properties under which sampled inputs will have high performance. Intuitively, model sensitivity could be reduced where the conditioning happens, which is however either surrogate- or search-specific. As a result, the conditioning is localized depending on how these out-of-distribution inputs and domain-specific properties are specified. It therefore remains unclear what impact such localized conditioning has on the overall model sensitivity. More importantly, it is also unclear whether in such approaches, the surrogate is

well-conditioned at the regions where the search might visit. Otherwise, it would be less effective if the surrogate is only well-conditioned in regions where the search rarely visits. In this view, the core issue here is the lack of a model-agnostic characterization of sensitivity, which does not depend on the specifics of either the search or surrogate models. Such characterization could play the role of a negotiating medium that coordinates the conditioning between the search and surrogate models.

This prompts two fundamental inquiries: (1) how to explicitly characterize and regulate the sensitivity of a surrogate model; and (2) whether such sensitivity-regulated approach can be agnostic of the search and surrogate model specifications, allowing it to be seamlessly integrated into both the search process and surrogate models to enhance the overall optimization performance. To shed light on these matters, our paper formalizes a model-agnostic sensitivity measure which can be optimized to coordinate the surrogate and search biases so that the surrogate is conditioned to be risk-averse wherever the search goes. This is substantiated with the following technical contributions:

**1.** We develop a novel concept of model sensitivity in terms of how often its output would change beyond a user-specified threshold under small model perturbations. Intuitively, if a model's output is sensitive to such perturbations, its prediction margin must be narrow and have high variance, which means the chance that it would fit well the unseen part of the oracle function is low. The developed sensitivity notion therefore provides a quantifiable risk assessment that can be exploited by both the surrogate and search models (Section 3).

**2.** We show that our model sensitivity measurement is optimizable and can be used as a regularizer for a diverse range of existing offline optimizers which either define their search based on derivatives of their surrogate models or couple both search and surrogate models in a single differentiable loss function. To enable this, we also develop a numerical algorithm to tractably and effectively optimize our sensitivity-informed regularizers (Section 4).

**3.** We demonstrate the empirical efficiency of the proposed regularization method on a wide variety of benchmark problems, which shows consistently that its synergistic performance boost on existing offline optimizers is significant. Overall, our results corroborate our earlier hypothesis that a model-agnostic characterization of sensitivity can help coordinate the conditioning between the search and surrogate models better, which is evident from the consistent performance improvement across many baselines and optimization tasks (Section 5).

**4.** For clarity, we also provide a concise review of the existing literature in Section 2.

## 2 RELATED WORK

In numerous material engineering fields, such as molecular structure, robot morphology, and protein design, the primary objective is to discover the optimal design that maximizes performance based on specific criteria. A significant challenge in addressing this problem arises from the fact that the relationship between a design and its performance is a black-box function, which requires expensive experiments or simulations to evaluate the performance output of each candidate input. Consequently, finding the optimal design is equivalent to optimizing a black-box function whose derivative information is not accessible.

Furthermore, sampling data from this black-box function also requires excessive laboring cost of conducting biophysical experiments, which makes existing derivative-free methods, such as random gradient estimation (Wang et al., 2018) or Bayesian optimization (Snoek et al., 2015), not economically viable as they often require sampling a large amount of data. This has inspired a new paradigm of offline optimization which learns an explicit model parameterization that explains well the relationship between input candidates and their corresponding experiment results in a past dataset.

To date, there have been several solutions proposed for offline optimization, which mostly focus on encoding some conservative preferences in either the search process or the (surrogate) training process. Such conservative preferences often aim to compensate for potential erratic function approximation so as to avoid false optimism during extrapolation. For example, Trabucco et al. (2021) forces the model to underestimate the output value of input candidates found during early iterations of gradient updates (deemed out-of-distribution), whereas Fu & Levine (2021) maximizes the normalized data likelihood to reduce uncertainty in out-of-distribution prediction. Yu et al. (2021) adopts techniques in model pre-training and adaptation to enforce a criteria of local smoothness.

Alternatively, Brookes et al. (2019) and Fannjiang & Listgarten (2020) focus instead on conditioning the search process. Under this paradigm, the search model is represented as a distribution conditioned on rare event of achieving high oracle performance, which is substantiated using different approaches. For instance, Brookes et al. (2019) models such conditioned distribution via an adversarial zero-sum game while Kumar & Levine (2020) learns an inverse mapping from the performance output to the input design using conditional generative adversarial network (Mirza & Osindero, 2014), from which design candidates performing at least as good as the example candidates in the offline dataset can be sampled. However, as mentioned previously, the implicit conditioning of these approaches is either search- or surrogate-specific, which does not coordinate well between the search and (surrogate) training processes. As such, the risk of following a search model will depend on how accurate the conditioning is at out-of-distribution inputs. This depends on the specific the conditioning algorithm that was adopted, which has neither been defined nor investigated.

## 3 A SENSITIVITY-GUIDED FRAMEWORK FOR OFFLINE OPTIMIZATION

In what follows, we will define the problem setting for offline optimization and introduce key notations (Section 3.1). We formalize the concept of model sensitivity (Section 3.2) and develop a sensitivity-informed regularizer for existing offline optimizers (Section 3.3). For clarity, an overview of our solution workflow is visualized in Figure 1.

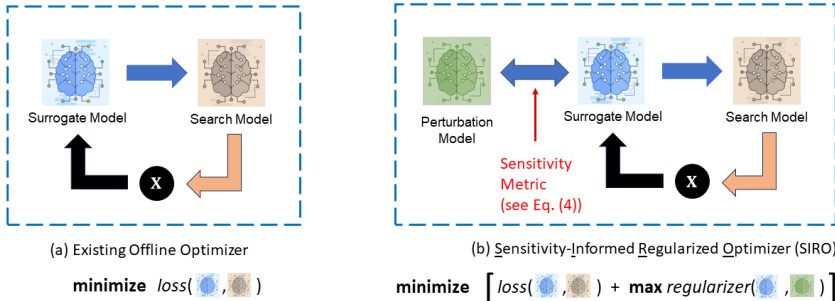

Figure 1: Workflows of (a) existing offline optimizers; and (b) our sensitivity-informed regularized optimizer (SIRO) which regulates the training workflow of existing offline optimizers with a new sensitivity metric – see Definition 1. Our regularizer is generic and can be applied to most existing offline optimizer workflows to boost their performance as demonstrated in Section 5.

### 3.1 PROBLEM SETTING AND NOTATIONS

A design problem is formulated as finding an optimal input or design $\mathbf{x}_* \in \mathfrak{X}$ that maximizes the output of an experiment or simulation process $g(\mathbf{x})$,

$$\mathbf{x}_* \triangleq \underset{\mathbf{x} \in \mathfrak{X}}{\arg\max} \; g(\mathbf{x}) . \tag{1}$$

As mentioned previously, we cannot access the black-box function $g(\mathbf{x})$ but we are provided with a set $\mathfrak{D}$ of $n$ training data points $(\mathbf{x}_i, z_i)_{i=1}^n$ such that $z_i = g(\mathbf{x}_i)$. This allows us to learn a surrogate $g(\mathbf{x}; \phi)$ of $g(\mathbf{x})$ via supervised learning,

$$\phi \triangleq \underset{\phi'}{\arg\min} \; \mathcal{L}(\phi'; \mathfrak{D}) \quad \text{where} \quad \mathcal{L}(\phi'; \mathfrak{D}) \triangleq \sum_{i=1}^n \text{err}\big(g(\mathbf{x}_i; \phi'), z_i\big) , \tag{2}$$

where $\phi'$ ($\phi$) denotes the surrogate parameterization and $\text{err}(z', z)$ denotes the loss of predicting $z'$ when the oracle value is $z$. For example, $\text{err}(z', z) = (z' - z)^2$ and $g(\mathbf{x}; \phi) = \phi^\top \mathbf{x}$. Once learned, $\phi$ is fixed and we can use $g(\mathbf{x}; \phi)$ as a surrogate to find (approximately) the optimal design,

$$\mathbf{x}_\phi \triangleq \underset{\mathbf{x} \in \mathfrak{X}}{\arg\max} \; g(\mathbf{x}; \phi) . \tag{3}$$

The quality of $\mathbf{x}_\phi$ is then defined as the (normalized) difference in oracle output between $\mathbf{x}_\phi$ and the oracle maximizer $\mathbf{x}_*$, $\mathfrak{C}(\mathbf{x}_\phi) = |g(\mathbf{x}_*) - g(\mathbf{x}_\phi)| \, / \, |g(\mathbf{x}_*) - \min_{\mathbf{x}} g(\mathbf{x})| \in (0, 1)$. Naively, if the surrogate $g(\mathbf{x}; \phi)$ is accurate over the entire input space $\mathfrak{X}$ then solving equation 3 is all we need. However, in most cases, $g(\mathbf{x}; \phi)$ is only accurate near the training data regime.

## 3.2 Sensitivity of Surrogate Model

Inspired by recent work in assessing model sensitivity (Stephenson et al., 2022; Tsai et al., 2021), the prediction of a model at a particular input $\mathbf{x}$ is considered sensitive if we can find a slightly perturbed variant of it that produces a significantly different prediction at $\mathbf{x}$. Following this intuition, we propose to measure the sensitivity of a model on the input space $\mathfrak{X}$ as the probability (over random perturbation) that the absolute difference between its expected output before and after perturbation is larger than a certain threshold. This is formalized below.

**Definition 1** *The $(\alpha, \omega)$-sensitivity of a model $g(\mathbf{x}; \phi)$ on the input space $\mathfrak{X}$[1] is defined as*

$$\mathcal{S}_\phi(\alpha, \omega) \triangleq \Pr_{\gamma \sim \mathbb{N}(\omega_\mu, \omega_\sigma^2 \mathbf{I})} \left( \left| \mathbb{E}_{\mathbf{x} \sim \mathfrak{X}} \Big[ g(\mathbf{x}; \phi + \gamma) \Big] - \mathbb{E}_{\mathbf{x} \sim \mathfrak{X}} \Big[ g(\mathbf{x}; \phi) \Big] \right| \geq \alpha \right) \quad (4)$$

*where $\omega = (\omega_\mu, \omega_\sigma)$ defines the parameter of the perturbation distribution, which is Gaussian.*

Intuitively, the above definition implies if there is a high chance that the expected output of a model $g(\mathbf{x}; \phi)$ would change significantly (larger than $\alpha$) within $\mathfrak{X}$ when its parameterization $\phi$ is perturbed by a certain amount of noise $\gamma$ controlled by $\omega$, then the model is sensitive. Hence, we want to condition the surrogate $g(\mathbf{x}; \phi)$ such that its induced sensitivity $\mathcal{S}_\phi(\alpha, \omega)$ is low. Otherwise, the larger $\mathcal{S}_\phi(\alpha, \omega)$ is, the more likely there exists a neighborhood of input at which the surrogate prediction is brittle against small perturbations of the model. This is established below.

**Lemma 1** *Let $\mathcal{S}_\phi(\alpha, \omega)$ defined via Definition 1. Suppose $\mathcal{S}_\phi(\alpha, \omega) \geq 1 - \delta$ with $\delta \in (0, 1)$. Then, with probability at least $1 - \delta$ over the space of random perturbation $\gamma \sim \mathbb{N}(\omega_\mu, \omega_\sigma^2 \mathbf{I})$, there exists $\mathbf{x} \in \mathfrak{X}$ such that $|g(\mathbf{x}; \phi + \gamma) - g(\mathbf{x}; \phi)| \geq \alpha$ and*

$$\forall \mathbf{x}' \in \mathfrak{X} : \left\| \mathbf{x}' - \mathbf{x} \right\| \leq \frac{\alpha}{4\mathfrak{L}_\phi} , \text{ we have } \left| g(\mathbf{x}'; \phi + \gamma) - g(\mathbf{x}'; \phi) \right| \geq \frac{\alpha}{2} \quad (5)$$

*where $\mathfrak{L}_\phi \triangleq \max_\gamma \mathfrak{L}_{\phi+\gamma}$ and $\mathfrak{L}_{\phi+\gamma}$ denotes the corresponding Lipschitz constant[2] of the surrogate $g(.; \phi + \gamma)$. A detailed derivation of this lemma can be found in Appendix A.*

This sensitivity measurement is however relative to a specific configuration of $(\alpha, \omega)$, which needs to be set meticulously. Otherwise, it is easy to see that $\mathcal{S}_\phi(\alpha, \omega)$ would quickly become vacuous (e.g., reaching the maximum value of one regardless of the choice of $\phi$) with decreasing values of $\alpha$ and increasing values of $(\omega_\mu, \omega_\sigma^2)$. Conversely, if $\alpha$ is too large while $(\omega_\mu, \omega_\sigma^2)$ are too small, $\mathcal{S}_\phi(\alpha, \omega)$ instead approaches zero regardless of the choice of $\phi$ and again, becomes vacuous.

Intuitively, this means the above sensitivity measure is only meaningful at the right range of values for $\alpha$ and $(\omega_\mu, \omega_\sigma^2)$ which needs to be determined via empirical observations. Following the practice in Trabucco et al. (2021), we conduct ablation studies in Section 5.3 to set the most robust, universal values (across all tasks) for $\alpha$ as well as the ranges on which $(\omega_\mu, \omega_\sigma^2)$ are optimized.

## 3.3 Sensitivity-Informed Regularizer

The previous discussion suggests that among surrogate candidates $g(\mathbf{x}; \phi)$ that fit equally well to the dataset, we would prefer one whose sensitivity $\mathcal{S}_\phi$ is smallest. Such surrogates tend to have prediction boundaries with (relatively) larger margins, which reduce the risk of being misguided by spurious predictions. Thus, suppose that the surrogate $g(\mathbf{x}; \phi)$ is fitted to offline data via minimizing a loss function $\mathcal{L}(\phi)$, we can regularize it via the following augmentation:

$$\phi = \arg\min_{\phi'} \left( \mathcal{L}(\phi') + \lambda \cdot \mathcal{S}_{\phi'}(\alpha, \omega) \right) \quad \text{where} \quad \omega = \arg\max_{\omega'} \mathcal{S}(\alpha, \omega') \quad (6)$$

where $\lambda > 0$ is a hyper-parameter regularizing between the two objectives: (1) fitting to offline data and (2) minimizing sensitivity. This features a bi-level optimization task, which can be relaxed into a minimax optimization task,

$$\phi = \arg\min_{\phi'} \left( \mathcal{L}(\phi') + \lambda \cdot \max_{\omega'} \mathcal{S}_{\phi'}(\alpha, \omega') \right) . \quad (7)$$

---

[1] $\mathfrak{X}$ is the entire input space that includes the example inputs in the offline dataset.
[2] $\mathfrak{L}$ is a Lipschitz constant of a function $g(\mathbf{x})$ if it is the smallest value for which $|g(\mathbf{x}) - g(\mathbf{x}')| \leq \mathfrak{L}\|\mathbf{x} - \mathbf{x}'\|$ for all pairs of input $(\mathbf{x}, \mathbf{x}')$.

The resulting formulation can now be solved approximately via alternating between (1) minimizing $\phi'$ while fixing $\omega'$ and (2) maximizing $\omega'$ while fixing $\phi'$. Intuitively, this process features a two-player game where one seeks to decrease the function value while the other seeks to increase it. The alternating optimization thus mimics a fictitious play that often finds an optimal equilibrium between the surrogate and the perturbation model. At this equilibrium, the surrogate has its sensitivity minimized in the worst case (against a most adversarial perturbation). This is in fact similar to the intuition behind Generative Adversarial Networks (GANs) (Goodfellow et al., 2014).

In addition, our regularization technique here is also agnostic to the specific choice of the loss function $\mathcal{L}(\phi')$. This makes our regularization technique synergistically amenable to a wider range of offline optimizers which could involve both the search and surrogate models. However, the main issue with this approach is that the mathematical characterization of $\mathcal{S}_\phi(\alpha, \omega)$ (see Definition 1) is not differentiable with respect to either $\omega$ or $\phi$, which prevents it from being optimized with gradient descent/ascent. This is a practical challenge which will be addressed next.

## 4 PRACTICAL ALGORITHM

To enable numerical optimization of the sensitivity measure $\mathcal{S}_\phi(\alpha, \omega)$ effectively, we will develop approximations of $\mathcal{S}_\phi(\alpha, \omega)$ that can be differentiated with respect to $\phi$ and $\omega$, respectively. These approximations are detailed below.

**Optimizing $\omega$.** First, given the sensitive threshold $\alpha$ and current surrogate estimate $\phi$, we define

$$\mathfrak{R}_\alpha(\phi) \quad \triangleq \quad \left\{ \gamma \;\middle|\; \left| \mathbb{E}\Big[g(\mathbf{x}; \phi + \gamma)\Big] - \mathbb{E}\Big[g(\mathbf{x}; \phi)\Big]\right| \;\geq\; \alpha \right\} . \tag{8}$$

Next, we observe that

$$
\begin{aligned}
\mathcal{S}_\phi(\alpha, \omega) &= \mathrm{Pr}_{\gamma \sim \mathbb{N}(\omega_\mu, \omega_\sigma^2 \mathbf{I})}\Big(\gamma \in \mathfrak{R}_\alpha(\phi)\Big) = \mathbb{E}_{\gamma \sim \mathbb{N}(\omega_\mu, \omega_\sigma^2 \mathbf{I})}\Big[\mathrm{Pr}\Big(\gamma \in \mathfrak{R}_\alpha(\phi) \;\Big|\; \gamma\Big)\Big] \\
&= \mathbb{E}_{\epsilon \sim \mathbb{N}(0, \mathbf{I})}\Big[\mathrm{Pr}\Big(\omega_\mu + \omega_\sigma \cdot \epsilon \in \mathfrak{R}_\alpha(\phi) \;\Big|\; \epsilon\Big)\Big] \quad \text{where we represent } \gamma = \omega_\mu + \omega_\sigma \cdot \epsilon \\
&\simeq \frac{1}{m} \sum_{i=1}^{m} \mathrm{Pr}\Big(\omega_\mu + \omega_\sigma \cdot \epsilon_i \in \mathfrak{R}_\alpha(\phi) \;\Big|\; \epsilon_i\Big) \simeq \frac{1}{m} \sum_{i=1}^{m} \Phi\Big(\gamma_i; \mathbf{w}\Big)
\end{aligned} \tag{9}
$$

where $\{\gamma_i = \omega_\mu + \omega_\sigma \cdot \epsilon_i\}_{i=1}^{m}$ and $\{\epsilon_i\}_{i=1}^{m}$ are identical and independent samples drawn from $\mathbb{N}(0, \mathbf{I})$ while $\Phi(\gamma_i; \mathbf{w})$ is a learnable neural net that predicts the probability that $\gamma_i \in \mathfrak{R}_\alpha(\phi)$.

We will refer to the last step in equation 9 above as the neural re-parameterization, which can approximate $\mathcal{S}_\phi(\alpha, \omega)$ arbitrarily closely given a sufficiently large number $m$ of samples and that the parameterization of $\Phi(\gamma; \mathbf{w})$ is sufficiently rich. Given a set of samples $(\kappa_i, \gamma_i)_{i=1}^{m}$ where $\kappa_i \triangleq \mathbb{I}(\gamma_i \in \mathfrak{R}_\alpha(\phi)) \in \{0, 1\}$, we can learn this neural re-parameterization via solving

$$\mathbf{w} = \underset{\mathbf{w}'}{\arg\max} \quad \Big[\Big(1 - \kappa_i\Big) \log \Big(1 - \Phi\Big(\gamma_i; \mathbf{w}'\Big)\Big) + \kappa_i \log \Phi\Big(\kappa_i; \mathbf{w}'\Big)\Big] \tag{10}$$

which is a standard logistic regression losses with parameterized bias $\Phi(\gamma; \mathbf{w})$ and training data $(\kappa_i, \gamma_i)_{i=1}^{m}$. Note that $\kappa_i = \mathbb{I}(\gamma_i \in \mathfrak{R}_\alpha(\phi))$ which is tractable. Once learned, we can take advantage of the differentiability of $\Phi(\gamma; \mathbf{w})$ to compute the gradient of $\mathcal{S}_\phi$ with respect to $\omega$:

$$\frac{\partial \mathcal{S}_\phi}{\partial \omega} \simeq \frac{1}{m} \sum_{i=1}^{m} \frac{\partial \Phi}{\partial \omega} = \frac{1}{m} \sum_{i=1}^{m} \left(\frac{\partial \Phi}{\partial \gamma_i} \cdot \frac{\partial \gamma_i}{\partial \omega}\right) . \tag{11}$$

Now recall that by the change of parameter, $\gamma_i = \omega_\mu + \omega_\sigma \cdot \epsilon_i$. Thus, $\partial \gamma_i / \partial \omega_\mu = 1$ and $\partial \gamma_i / \partial \omega_\sigma = \epsilon_i$. As such, replacing $\omega$, respectively, with $\omega_\mu$ and $\omega_\sigma$ in equation 11 produces[3]

$$\frac{\partial \mathcal{S}_\phi}{\partial \omega_\mu} = \frac{1}{m} \sum_{i=1}^{m} \frac{\partial \Phi}{\partial \gamma_i} \quad \text{and} \quad \frac{\partial \mathcal{S}_\phi}{\partial \omega_\sigma} = \frac{1}{m} \sum_{i=1}^{m} \epsilon_i \cdot \frac{\partial \Phi}{\partial \gamma_i} . \tag{12}$$

---

[3]Note that we abuse the notation a bit here to treat $\epsilon_i$ as a scalar while it should be a random vector. This is not an issue since the covariance matrix of its distribution $\mathbb{N}(0, \mathbf{I})$ is diagonal, which implies components of the random vector are i.i.d and hence, equation 12 applies separately to each such scalar component.

---

**Algorithm 1** `SIRO` (Sensitivity-Informed Regularization for Offline Optimization)

---

**Input:** offline data $\mathfrak{D} = \{(\mathbf{x}_i, z_i)\}_{i=1}^n$; initial surrogate model $g(\mathbf{x}; \phi)$; initial perturbation parameters $\omega = (\omega_\mu, \omega_\sigma^2)$; no. $m$ of sampled perturbations $\gamma \sim \mathbb{N}(\omega_\mu, \omega_\sigma^2 \mathbf{I})$; no. of iterations $\tau$; sensitivity threshold $\alpha$; learning rates $(\eta_\omega, \eta_\phi)$

---

1: initialize $\phi^{(1)} \leftarrow \phi$ and $\omega^{(1)} \leftarrow \omega$
2: **for** $t \leftarrow 1 : \tau$ **do**
3:      sample $\{\gamma_i\}_{i=1}^m$ where $\gamma_i = \omega_\mu^{(t)} + \omega_\sigma^{(t)} \cdot \epsilon_i$ where $\{\epsilon_i\}_{i=1}^m \sim \mathbb{N}(0, \mathbf{I})$
4:      compute $\nabla_\phi h(\phi) = \mathbb{E}_{\mathbf{x} \sim \mathfrak{D}}\left[\nabla_\phi g(\mathbf{x}; \phi)\right]$ at $\phi = \phi^{(t)}$ – see Eq. 14
5:      **for** $i \leftarrow 1 : m$ **do**
6:          $\kappa_i \leftarrow \mathbb{I}\left(\left|\nabla_\phi h(\phi)^\top \gamma_i\right| > \alpha\right)$ – label whether or not $\gamma_i \in \mathfrak{R}_\alpha(\phi)$ – see Eq. 8
7:      learn neural re-parameterization $\Phi(\gamma; \mathbf{w})$ with dataset $\{(\gamma_i, \kappa_i)\}_{i=1}^m$ – see Eq. 10
8:      **optimizing $\omega$:**
9:      compute $\nabla_{\omega_\mu} \mathcal{S}_\phi \leftarrow m^{-1}\left(\sum_{i=1}^m \nabla_\gamma \Phi(\gamma_i)\right)$ via Eq. 12
10:     compute $\nabla_{\omega_\sigma} \mathcal{S}_\phi \leftarrow m^{-1}\left(\sum_{i=1}^m \epsilon_i \cdot \nabla_\gamma \Phi(\gamma_i)\right)$ via Eq. 12
11:     update $\omega_\mu^{(t+1)} \leftarrow \omega_\mu^{(t)} + \eta_\omega \cdot \nabla_{\omega_\mu} \mathcal{S}_\phi$ and $\omega_\sigma^{(t+1)} \leftarrow \omega_\sigma^{(t)} + \eta_\omega \cdot \nabla_{\omega_\sigma} \mathcal{S}_\phi$
12:     **optimizing $\phi$:**
13:     compute $\mathcal{S}_\phi^+(\alpha, \omega) = m^{-1} \sum_{i=1}^m \left[\min\left(1, \left(\left(\nabla_\phi h(\phi)^\top \gamma_i\right)^2 / \alpha^2\right)\right)\right]$ – Eq. 13 and Eq. 14
14:     update $\phi^{(t+1)} \leftarrow \phi^{(t)} + \eta_\phi \cdot \left(\nabla_\phi \mathcal{L}(\phi; \mathfrak{D}) + \lambda \cdot \nabla_\phi \mathcal{S}_\phi^+(\alpha, \omega)\right)$ at $\phi = \phi^{(t)}$ and $\omega = \omega^{(t+1)}$
     **return** learned surrogate $g\left(\mathbf{x}; \phi^{(\tau+1)}\right)$

---

Hence, equation 12 enables numerical maximization of $\omega$ via gradient ascent.

**Optimizing $\phi$.** To optimize $\phi$, we need another approximation since the neural re-parameterization above is still not differentiable with respect to $\phi$. This is developed via the following lemma.

**Lemma 2** *Let $\mathcal{S}_\phi(\alpha, \omega)$ defined via Definition 1. We have $\mathcal{S}_\phi(\alpha, \omega) \leq \mathcal{S}_\phi^+(\alpha, \omega)$ where*

$$\mathcal{S}_\phi^+(\alpha, \omega) = \mathbb{E}_{\gamma \sim \mathbb{N}(\omega_\mu, \omega_\sigma^2 \mathbb{I})}\left[\min\left(1, \frac{1}{\alpha^2} \cdot \left(\mathbb{E}_{\mathbf{x} \in \mathfrak{X}}\left[g(\mathbf{x}; \phi + \gamma) - g(\mathbf{x}; \phi)\right]\right)^2\right)\right] \quad (13)$$

*A detailed derivation of this lemma is provided in Appendix B.*

Lemma 2 thus establishes an upper bound for $\mathcal{S}_\phi(\alpha, \omega)$, which is now differentiable with respect to $\phi$. Note that while minimizing $\phi$, $\omega = (\omega_\mu, \omega_\sigma^2)$ is fixed and hence, the expectation does not involve parameters that need to be differentiated. Thus, the derivative operator can be pushed inside the expectation, which is differentiable with respect to $\phi$.

However, one minor detail here is that computing $\mathcal{S}_\phi^+(\alpha, \omega)$ still requires computing the expected output difference for each sampled $\gamma$. This operation is not vectorizable and cannot take advantage of the GPU compute infrastructure. To sidestep this final hurdle, we propose to approximate $h(\phi+\gamma) \triangleq \mathbb{E}[g(\mathbf{x}; \phi + \gamma)]$ with a first-order Taylor expansion around $\phi$. That is,

$$\mathbb{E}[g(\mathbf{x}; \phi + \gamma)] - \mathbb{E}[g(\mathbf{x}; \phi)] = h(\phi + \gamma) - h(\phi) \simeq \nabla h(\phi)^\top \gamma . \quad (14)$$

This is in forms of a matrix multiplication involving $\gamma$, which is now vectorizable.

**Full Algorithm.** Putting the above together, we have the following implementable pseudo-code of our algorithm represented in Algorithm 1 below.

## 5 EXPERIMENTS

This section evaluates the effectiveness of our proposed regularizer (`SIRO`) on boosting the performance of existing state-of-the-art offline optimizers. Our empirical studies adopt the benchmark tasks from Design-Bench (Trabucco et al., 2022) and its widely recognized baseline algorithms and evaluation protocol (Section 5.1). All empirical results and analyses are reported in Section 5.2.

## 5.1 Benchmarks, Baselines and Evaluation Methodology

**1. Benchmark Tasks.** Our empirical evaluations are conducted on 6 real-world tasks from Design-Bench (Trabucco et al., 2022), spanning both discrete and continuous domains.

The discrete tasks include **TF-Bind-8**, **TF-Bind-10**, and **ChEMBL** where **TF-Bind-8** and **TF-Bind-10** (Barrera et al., 2016) involve discovering DNA sequences with high binding affinity to a specific transcription factor (SIX6 REF R1) with sequence lengths 8 and 10, respectively; and **ChEMBL** is derived from a drug property database (Gaulton et al., 2012) and requires optimizing a molecule for a high MCHC value when paired with assay CHEMBL3885882.

The continuous tasks include **Ant Morphology** (Brockman et al., 2016), **D'Kitty Morphology** (Ahn et al., 2020), and **Superconductors** (Brookes et al., 2019). In **Ant Morphology** and the **D'Kitty Morphology** task, we optimize the physical structure of a simulated robot Ant from OpenAI Gym (Brockman et al., 2016) and the D'Kitty robot from ROBEL (Ahn et al., 2020). The **Superconductor** task is about designing superconductor molecules that have the highest critical temperature.

**2. Baselines.** To assess how our proposed regularizer SIRO influences the performance of established baseline algorithms, we selected 10 widely recognized offline optimizers for comparison. These include **BO-qEI** (Trabucco et al., 2022), **CbAS** (Brookes et al., 2019), **RoMA** (Yu et al., 2021), **CMA-ES** (Hansen), **COMs** (Trabucco et al., 2021), **MINs** (Kumar & Levine, 2020), **REINFORCE** (Williams, 1992), and 3 variants of gradient ascent (**GA**, **ENS-MIN**, **ENS-MEAN**) which correspond to the vanilla gradient ascent, the min ensemble of gradient ascent, and the mean ensemble of gradient ascent.

**3. Evaluation Protocol.** For each baseline algorithm, we configure it with its corresponding best hyperparameters specified in (Trabucco et al., 2022). To provide a comprehensive evaluation of each algorithm's performance, we adhere to the recommended approach in (Trabucco et al., 2022), which requires each method to generate $K = 128$ optimized design candidates, which are then evaluated using the oracle function. The evaluated performance of these candidates are then sorted in increasing order from which performance at 50-th and 75-th and 100-th percentile levels are reported. All reported performance are averaged over 8 independent runs.

**4. Hyperparameter Configuration.**

Our proposed regularizer SIRO also has additional hyperparameters $(\alpha, \omega)$ as highlighted previously in Section 3.2. In particular, $\omega = (\omega_\mu, \omega_\sigma^2)$ is a tuple of learnable parameters that define the (adversarial) perturbation distribution $\mathbb{N}(\omega_\mu, \omega_\sigma^2 \mathbf{I})$, which is used to measure the sensitivity of the regularizer. However, the perturbation is supposed to be in the low-noise regime which requires the range of values for those parameters to be set so that the perturbation will not dominate the surrogate's parameters. Otherwise, our sensitivity measure will become vacuous. We have conducted ablation studies in Section 5.3 to find the most appropriate ranges for these parameters, which appears to be $[-10^{-3}, 10^{-3}]$ for $\omega_\mu$ and $[10^{-5}, 10^{-2}]$ for $\omega_\sigma$. We use the above ranges in all experiments, in which $\omega_\mu$ and $\omega_\sigma^2$ are initialized to 0 and $10^{-3}$, respectively. Likewise, for the sensitivity threshold, our ablation studies observe the impact of several values of $\alpha$ on the performance and find that $\alpha = 0.1$ is the best universal value for SIRO. In addition, we set the weight of the regularizer $\lambda$ to $10^{-3}$, the no. $m$ of perturbation sample per iteration to 100 and the learning rates $\eta_\omega = 10^{-2}$, $\eta_\phi = 10^{-3}$. We find that this configuration is universally robust across all benchmark tasks. $\Phi$ is a neural network with one hidden layer comprising two hidden units and one output layer.

## 5.2 Results and Discussion

This section reported the percentage of improvement over baseline performance achieved by SIRO when it is applied to an existing baseline. We have evaluated this at 50-th, 75-th and 100-th percentile levels. However, due to limited space, we only report result of the 100-th percentile level in the main text. The other results are instead deferred to Appendix C.3.

**Results on Continuous Tasks.** The first part (the first 3 column) of Table 1 shows that among 30 cases (across 10 baselines and 3 tasks), incorporating the SIRO regularizer improves the baseline performance positively up to $9.4\%$. There is only one instance where SIRO decreases the performance but the decrease is only $0.2\%$, which is almost negligible.

Table 1: Percentage of performance improvement achieved by SIRO across all tasks and baselines at the 100-**th percentile** level. **P** denote the achieved normalized performance while **G** denote SIRO's percentage of performance gain over the baseline performance.

| Algorithms | | Continuous Tasks | | | | | | Discrete Tasks | | | | | |
|---|---|---|---|---|---|---|---|---|---|---|---|---|---|
| | | Ant Morphology | | D'Kitty Morphology | | Superconductor | | TF Bind 8 | | TF Bind 10 | | ChEMBL | |
| | | P | G | P | G | P | G | P | G | P | G | P | G |
| $\mathcal{D}$(best) | | 0.565 | | 0.884 | | 0.400 | | 0.439 | | 0.467 | | 0.605 | |
| CbAS | Base | 0.856 ± 0.029 | | 0.895 ± 0.011 | | 0.480 ± 0.038 | | 0.911 ± 0.034 | | 0.615 ± 0.031 | | 0.636 ± 0.005 | |
| | SIRO | 0.861 ± 0.032 | +0.5% | 0.907 ± 0.012 | +1.2% | 0.485 ± 0.025 | +0.5% | 0.919 ± 0.054 | +0.8% | 0.656 ± 0.042 | +4.1% | 0.637 ± 0.010 | +0.1% |
| BO-qEI | Base | 0.812 ± 0.000 | | 0.896 ± 0.000 | | 0.394 ± 0.048 | | 0.779 ± 0.125 | | 0.692 ± 0.126 | | 0.659 ± 0.023 | |
| | SIRO | 0.812 ± 0.000 | +0.0% | 0.896 ± 0.000 | +0.0% | 0.464 ± 0.013 | +7.0% | 0.802 ± 0.069 | +2.3% | 0.692 ± 0.126 | +0.0% | 0.688 ± 0.000 | +2.9% |
| CMA-ES | Base | 1.915 ± 0.909 | | 0.723 ± 0.001 | | 0.481 ± 0.026 | | 0.944 ± 0.035 | | 0.676 ± 0.039 | | 0.633 ± 0.000 | |
| | SIRO | 2.009 ± 1.540 | +9.4% | 0.725 ± 0.002 | +0.2% | 0.482 ± 0.024 | +0.1% | 0.941 ± 0.029 | -0.3% | 0.672 ± 0.063 | -0.4% | 0.633 ± 0.000 | +0.0% |
| GA | Base | 0.299 ± 0.037 | | 0.871 ± 0.012 | | 0.506 ± 0.008 | | 0.980 ± 0.015 | | 0.647 ± 0.029 | | 0.640 ± 0.010 | |
| | SIRO | 0.314 ± 0.034 | +1.5% | 0.883 ± 0.012 | +1.2% | 0.515 ± 0.014 | +0.9% | 0.986 ± 0.007 | +0.6% | 0.658 ± 0.072 | +1.1% | 0.646 ± 0.002 | +0.6% |
| ENS-MIN | Base | 0.399 ± 0.077 | | 0.892 ± 0.010 | | 0.501 ± 0.013 | | 0.986 ± 0.006 | | 0.642 ± 0.025 | | 0.653 ± 0.018 | |
| | SIRO | 0.472 ± 0.110 | +7.3% | 0.893 ± 0.009 | +0.1% | 0.504 ± 0.010 | +0.3% | 0.989 ± 0.007 | +0.3% | 0.662 ± 0.038 | +2.0% | 0.662 ± 0.007 | +0.9% |
| ENS-MEAN | Base | 0.403 ± 0.045 | | 0.897 ± 0.009 | | 0.510 ± 0.012 | | 0.984 ± 0.007 | | 0.628 ± 0.028 | | 0.653 ± 0.014 | |
| | SIRO | 0.412 ± 0.094 | +0.9% | 0.897 ± 0.005 | +0.0% | 0.511 ± 0.015 | +0.1% | 0.986 ± 0.007 | +0.2% | 0.641 ± 0.032 | +1.3% | 0.662 ± 0.009 | +0.9% |
| REINFORCE | Base | 0.253 ± 0.047 | | 0.674 ± 0.138 | | 0.481 ± 0.015 | | 0.929 ± 0.031 | | 0.664 ± 0.061 | | 0.634 ± 0.002 | |
| | SIRO | 0.278 ± 0.014 | +2.5% | 0.732 ± 0.003 | +5.8% | 0.483 ± 0.007 | +0.2% | 0.943 ± 0.029 | +1.4% | 1.000 ± 0.000 | +33.6% | 0.639 ± 0.010 | +0.5% |
| MINs | Base | 0.906 ± 0.019 | | 0.944 ± 0.009 | | 0.461 ± 0.027 | | 0.907 ± 0.051 | | 0.636 ± 0.039 | | 0.633 ± 0.000 | |
| | SIRO | 0.924 ± 0.016 | +1.8% | 0.942 ± 0.008 | -0.2% | 0.476 ± 0.023 | +1.5% | 0.938 ± 0.053 | +3.1% | 0.644 ± 0.047 | +0.8% | 0.634 ± 0.003 | +0.1% |
| COMs | Base | 0.896 ± 0.024 | | 0.937 ± 0.012 | | 0.483 ± 0.026 | | 0.946 ± 0.035 | | 0.628 ± 0.044 | | 0.633 ± 0.000 | |
| | SIRO | 0.918 ± 0.025 | +2.2% | 0.942 ± 0.011 | +0.5% | 0.486 ± 0.030 | +0.3% | 0.954 ± 0.020 | +0.8% | 0.664 ± 0.038 | +3.6% | 0.638 ± 0.009 | +0.5% |
| RoMA | Base | 0.553 ± 0.094 | | 0.834 ± 0.017 | | 0.489 ± 0.022 | | 0.665 ± 0.000 | | 0.553 ± 0.000 | | 0.633 ± 0.000 | |
| | SIRO | 0.606 ± 0.059 | +5.3% | 0.836 ± 0.017 | +0.2% | 0.498 ± 0.023 | +0.9% | 0.665 ± 0.000 | +0.0% | 0.553 ± 0.000 | +0.0% | 0.633 ± 0.000 | +0.0% |

Moreover, in some cases, even when SIRO only maintains similar performance as the baseline, it still helps reduce the performance variance, as demonstrated by a reduction from 0.9% to 0.5% with **ENS-MEAN** baseline on the **D'kitty Morphology** task. In addition, for certain cases, SIRO also helps establish new SOTA performance. For example, in the **Ant Morphology** task where the SOTA baseline is represented by **CMA-ES** achieving 191.5%, incorporating SIRO elevates its performance by 200.9%, setting a new SOTA performance.

**Results on Discrete Tasks.** The second part (last 3 columns) of Table 1 illustrates the impact of our regularizer SIRO on the performance of baseline algorithms in 3 discrete domains (**TF-BIND-8**, **TF-BIND-10**, and **ChEMBL**). It is observed that in most cases (28/30), SIRO enhances the baseline algorithms' performance significantly up to 33.6%.

For the two instances in which SIRO decreases the performance, the amount of decrease is (relatively) much milder, ranging between 0.3% and 0.4%. Furthermore, on a closer look, it is also observable that for certain cases, the incorporation of SIRO also help elevate the state-of-the-art (SOTA) results (in addition to improving the baseline performance). For example, on **TF-BIND-8** and **ChEMBL**, the original SOTA performance (0.986 and 0.659) are achieved by **ENS-MIN** and **BO-qEI**, respectively. These are further elevated to 0.989 and 0.688 when SIRO is added to regulate the loss function of **ENS-MIN** and **BO-qEI**, thus establishing new SOTA.

Overall, our observations suggests that SIRO demonstrates consistently a high probability (81.67% = 49/60 cases) of improving baseline performance. On average, it leads to an improvement of approximately 2.35%, with a notable peak improvement of 33.6%. Conversely, SIRO also carries a relatively much lower probability (5.0% = 3/60 cases) of decreasing baseline performance. When such cases occur, the average performance decrease is at most 0.3%, which is almost negligible. Code for reproducing our results is at https://github.com/abcdefghhgfedcba/SIRO

## 5.3 ABLATION EXPERIMENTS

This section presents additional experiments to examine the sensitivity of two representative baselines, **COMs** and **GA** (regularized with SIRO) to changes in the number of gradient ascent steps performed during optimization. Furthermore, we also conduct ablation studies to investigate the effects of specific hyperparameters of SIRO (see Algorithm 1), including the sensitivity threshold $\alpha$, the no. of sampled perturbation $m$, and the effective value ranges for $\omega$, on the performance. Our studies are mainly conducted on two tasks, **SUPERCONDUCTOR** and **TF-BIND-8**.

**SIRO enhances stability of COMs and gradient ascent (GA).** Figure 2a depicts the performance variation of two baseline algorithms, **COMs**, and **GA**, with and without our SIRO regularizer. It

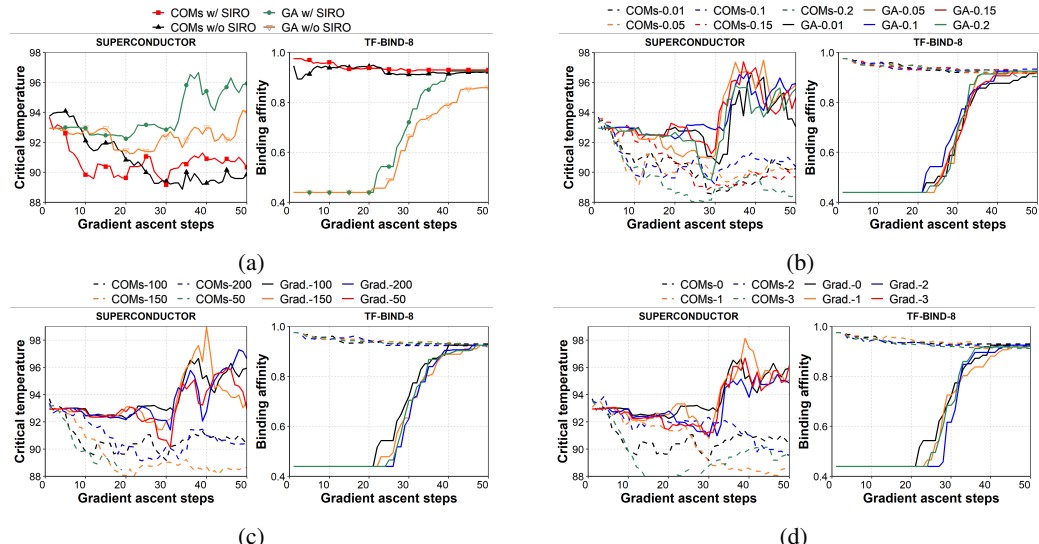

Figure 2: Plots of performance variation of **COMS** and **GA** (regularized by SIRO) to changes in (a) the no. of gradient ascent steps during optimization; (b) values of the sensitivity threshold $\alpha$; (c) changes in the no. of perturbation samples $m$; and (d) changes in value ranges of bound $([-\omega_{\mu_b}, \omega_{\mu_b}], [\omega_{\sigma_l}, \omega_{\sigma_u}])$ for parameter $\omega$ of the perturbation distribution. These are $([-10^{-3}, 10^{-3}], [10^{-5}, 10^{-2}])$, $([-10^{-3}, 10^{-3}], [10^{-6}, 10^{-3}])$, $([-10^{-2}, 10^{-2}], [10^{-5}, 10^{-2}])$, and $([-10^{-2}, 10^{-2}], [10^{-6}, 10^{-3}])$ which are indexed with 0, 1, 2, 3 in this figure.

is observed that initially these baselines outperform their SIRO-enhanced counterparts, but as the number of optimization steps increases, their performance starts to lag behind. This suggests that SIRO will become increasingly beneficial for improving the baseline performance in the latter stages of the optimization process.

**Choosing the sensitivity threshold $\alpha$ – see Definition 1**. Figure 2b visualizes how the performance of baselines regularized with SIRO is influenced by varying the value of $\alpha$. The results indicate that using either an excessively low or high value for $\alpha$ will impact the performance negatively. In all cases, the results suggest that a universal value of 0.1 for $\alpha$ tend to generate consistent and effective performance across all tasks.

**Choosing the no. $m$ of perturbation samples.** Figure 2c visualizes how the performance of baselines regularized with SIRO is influenced by varying the number of perturbation samples drawn from $\mathbb{N}(\omega_\mu, \omega_\sigma^2 \mathbf{I})$. It is observed that with more perturbation samples, the SIRO-regularized baseline achieves higher performance gain but also incurs more compute expense. As the performance gain beyond $m = 100$ appears marginal, we choose $m = 100$ in all our experiments.

**Choosing value ranges for $\omega$.** As we mentioned previously in Section 3.2, large values for $\omega$ can make the sensitivity measure vacuous because by definition, sensitivity characterizes changes under slight perturbation of the model weight. To find this appropriate range, we plot the performance of the **GA** and **COMS** baselines regularized with SIRO with respect to a set of potential ranges $([-\omega_{\mu_b}, \omega_{\mu_b}], [\omega_{\sigma_l}, \omega_{\sigma_u}])$ for $\omega_\mu$ and $\omega_\sigma$ in Figure 2d. The results indicate that using an excessively low or high range of values will impact the performance negatively. Overall, the empirical results suggest that $[-10^{-3}, 10^{-3}]$ and $[10^{-5}, 10^{-2}]$ are best value ranges for $\omega_\mu$ for $\omega_\sigma$, respectively.

## 6 CONCLUSION

This paper formalized the concept of model sensitivity in offline optimization, which inspires a new sensitivity-informed regularization that works synergistically with numerous existing approaches to boost their performance. As such, our contribution stands as an essential addition to the existing body of research in this field, providing a versatile and effective performance booster for a wide range of offline optimizers. This is extensively demonstrated on a diverse task benchmark. In addition, we believe the developed principles can also be adapted to related disciplines such as safe Bayesian optimization (BO) or safe reinforcement learning (RL) in online interactive learning scenarios, which are potential follow-up of our current work.

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

## A    APPENDIX A: DERIVATION OF LEMMA 1

The derivation of Lemma 1 goes as follows. First, note that:

$$\left| \mathbb{E}_{\mathbf{x} \sim \mathfrak{X}} \Big[ g(\mathbf{x}; \phi + \gamma) \Big] - \mathbb{E}_{\mathbf{x} \sim \mathfrak{X}} \Big[ g(\mathbf{x}; \phi) \Big] \right| \leq \mathbb{E}_{\mathbf{x} \sim \mathfrak{X}} \Big| g(\mathbf{x}; \phi + \gamma) - g(\mathbf{x}; \phi) \Big| \tag{15}$$

$$\leq \max_{\mathbf{x} \in \mathfrak{X}} \Big| g(\mathbf{x}; \phi + \gamma) - g(\mathbf{x}; \phi) \Big| \tag{16}$$

This implies the probability that the LHS is larger than $\alpha$ is smaller than the probability that the RHS is larger than $\alpha$. Hence,

$$1 - \delta \leq \mathcal{S}_\phi(\alpha, \omega) \leq \Pr_{\gamma \sim \mathbb{N}(\omega_\mu, \omega_\sigma^2 \mathbf{I})} \left\{ \max_{\mathbf{x} \in \mathfrak{X}} \Big| g(\mathbf{x}; \phi + \gamma) - g(\mathbf{x}; \phi) \Big| \geq \alpha \right\} \tag{17}$$

which implies with probability at least $1 - \alpha$, there exists an input at which the prediction might change by more than $\alpha$ due to a slight perturbation. This establishes the first part of Lemma 1.

Now, let this input be $\mathbf{x}$. Then, with probability at least $1 - \delta$:

$$\alpha \leq \big| g(\mathbf{x}; \phi + \gamma) - g(\mathbf{x}; \phi) \big| \leq \big| g(\mathbf{x}; \phi + \gamma) - g(\mathbf{x}'; \phi + \gamma) \big|$$
$$+ \big| g(\mathbf{x}'; \phi + \gamma) - g(\mathbf{x}'; \phi) \big| + \big| g(\mathbf{x}'; \phi) - g(\mathbf{x}; \phi) \big|$$
$$\leq 2\mathfrak{L}_\phi \| \mathbf{x} - \mathbf{x}' \| + \big| g(\mathbf{x}'; \phi + \gamma) - g(\mathbf{x}'; \phi) \big| \tag{18}$$

is true for any $\mathbf{x}' \in \mathfrak{X}$. To see this, the first step in equation 18 above follows from the triangle inequality and the second step follows from the definition of Lipschitz constant in Lemma 1. Now, if $\mathbf{x}'$ is within a $(\alpha/4\mathfrak{L}_\phi)$-ball centered at $\mathbf{x}$, we have $\| \mathbf{x} - \mathbf{x}' \| \leq \alpha/(4\mathfrak{L}_\phi)$.

Plugging this into equation 18 leads to:

$$\alpha \leq 2\mathfrak{L}_\phi \| \mathbf{x} - \mathbf{x}' \| + \big| g(\mathbf{x}'; \phi + \gamma) - g(\mathbf{x}'; \phi) \big|$$
$$\leq 2\mathfrak{L}_\phi \cdot \frac{\alpha}{4\mathfrak{L}_\phi} + \big| g(\mathbf{x}'; \phi + \gamma) - g(\mathbf{x}'; \phi) \big| = \frac{\alpha}{2} + \big| g(\mathbf{x}'; \phi + \gamma) - g(\mathbf{x}'; \phi) \big| \tag{19}$$

which implies $|g(\mathbf{x}'; \phi + \gamma) - g(\mathbf{x}'; \phi)| \geq \alpha - \alpha/2 = \alpha/2$. Thus, with probability $1 - \delta$, this is true for all $\mathbf{x}'$ in the $(\alpha/4\mathfrak{L}_\phi)$-ball centered at $\mathbf{x}$. The second part of Lemma 1 has been derived.

## B    APPENDIX B: DERIVATION OF LEMMA 2

The derivation of Lemma 2 goes as follows. First, note that:

$$\mathcal{S}_\phi(\alpha, \omega) \triangleq \Pr_{\gamma \sim \mathbb{N}(\omega_\mu, \omega_\sigma^2 \mathbf{I})} \left( \left| \mathbb{E}_{\mathbf{x} \sim \mathfrak{X}} \Big[ g(\mathbf{x}; \phi + \gamma) \Big] - \mathbb{E}_{\mathbf{x} \sim \mathfrak{X}} \Big[ g(\mathbf{x}; \phi) \Big] \right| \geq \alpha \right) \tag{20}$$

$$= \mathbb{E}_{\gamma \sim \mathbb{N}(\omega_\mu, \omega_\sigma^2 \mathbf{I})} \left[ \mathbb{I} \left( \left| \mathbb{E}_{\mathbf{x} \sim \mathfrak{X}} \Big[ g(\mathbf{x}; \phi + \gamma) \Big] - \mathbb{E}_{\mathbf{x} \sim \mathfrak{X}} \Big[ g(\mathbf{x}; \phi) \Big] \right| \geq \alpha \right) \right] \tag{21}$$

On the other hand, we have

$$A = \mathbb{I} \left( \left| \mathbb{E}_{\mathbf{x} \sim \mathfrak{X}} \Big[ g(\mathbf{x}; \phi + \gamma) \Big] - \mathbb{E}_{\mathbf{x} \sim \mathfrak{X}} \Big[ g(\mathbf{x}; \phi) \Big] \right| \geq \alpha \right) \tag{22}$$

$$< \frac{1}{\alpha^n} \cdot \left| \mathbb{E}_{\mathbf{x} \sim \mathfrak{X}} \Big[ g(\mathbf{x}; \phi + \gamma) \Big] - \mathbb{E}_{\mathbf{x} \sim \mathfrak{X}} \Big[ g(\mathbf{x}; \phi) \Big] \right|^n, \tag{23}$$

which is true for any $\gamma \sim \mathbb{N}(\omega_\mu, \omega_\sigma^2 \mathbf{I})$ and $\alpha > 0$. Thus, choosing $n = 2$ results in a differentiable (with respect to $\phi$) function that upper bounds the regularizer as follows:

$$\mathcal{S}_\phi(\alpha, \omega) < \frac{1}{\alpha^2} \cdot \mathbb{E}_{\gamma \sim \mathbb{N}(\omega_\mu, \omega_\sigma^2 \mathbf{I})} \left[ \left| \mathbb{E}_{\mathbf{x} \sim \mathfrak{X}} \Big[ g(\mathbf{x}; \phi + \gamma) \Big] - \mathbb{E}_{\mathbf{x} \sim \mathfrak{X}} \Big[ g(\mathbf{x}; \phi) \Big] \right|^2 \right] \tag{24}$$

However, when the difference between the surrogate model's prediction and the perturbed model's prediction is considerably larger than alpha, the upper bound will be significantly higher than the

original value, leading to reduced reliability of the upper bound. To address this issue, we employ the minimum function to set a threshold on the disparity between the upper bound regularizer and the original regularizer. By using the min function, we aim to reduce the difference between the upper bound regularizer and the original one, without affecting the computation of gradients since the min function is differentiable. Hence,

$$
\mathcal{S}_\phi(\alpha, \omega) \quad < \quad \mathop{\mathbb{E}}_{\gamma \sim \mathbb{N}(\omega_\mu, \omega_\sigma^2 \mathbf{I})} \left[ \min \left( 1, \frac{1}{\alpha^2} \cdot \left| \mathbb{E}_{\mathbf{x} \sim \mathfrak{X}} \left[ g(\mathbf{x}; \phi + \gamma) \right] - \mathbb{E}_{\mathbf{x} \sim \mathfrak{X}} \left[ g(\mathbf{x}; \phi) \right] \right|^2 \right) \right] \tag{25}
$$

$$
\triangleq \quad \mathcal{S}_\phi^+(\alpha, \omega) \tag{26}
$$

To evaluate the tightness of Lemma 2, we undertook a minor experiment to calculate the disparity between $S_\phi$ and $S_\phi^+$, i.e., $|S_\phi - S_\phi^+|$. The findings of this experiment are depicted in Figure 3c, where we present the average and standard deviation across batch data during 50 epochs of training Gradient ascent algorithm (GA) on two tasks: Superconductor and TF-BIND-8. The illustration reveals that the greatest divergence is approximately 7%, which is sufficiently minimal to affirm the accuracy of Lemma 2.

## C  APPENDIX C: ADDITIONAL EXPERIMENT RESULTS

### C.1  COMPARISION WITH OTHER REGULARIZATION METHODS

In the realm of machine learning, traditional $L1$ and $L2$-norm regularization techniques are known to prevent overfitting during model training. To evaluate their effectiveness, we carried out a compact experiment to juxtapose these methods with our regularization approach. This comparison involves examining the objective values of 128 solution designs through 50 steps of Gradient Ascent in two distinct tasks: Superconductor and TF-BIND-8. Figure 3a indicates that while $L1$ and $L2$ norms improve upon the baseline solution, they do not surpass the performance of our regularization technique. This is expected since `SIRO` is specifically designed to condition the output behavior of the model against adversarial perturbation while $L1$ and $L2$ only generically penalize models with high complexity, measured by the $L1$ and $L2$ norms of their parameters.

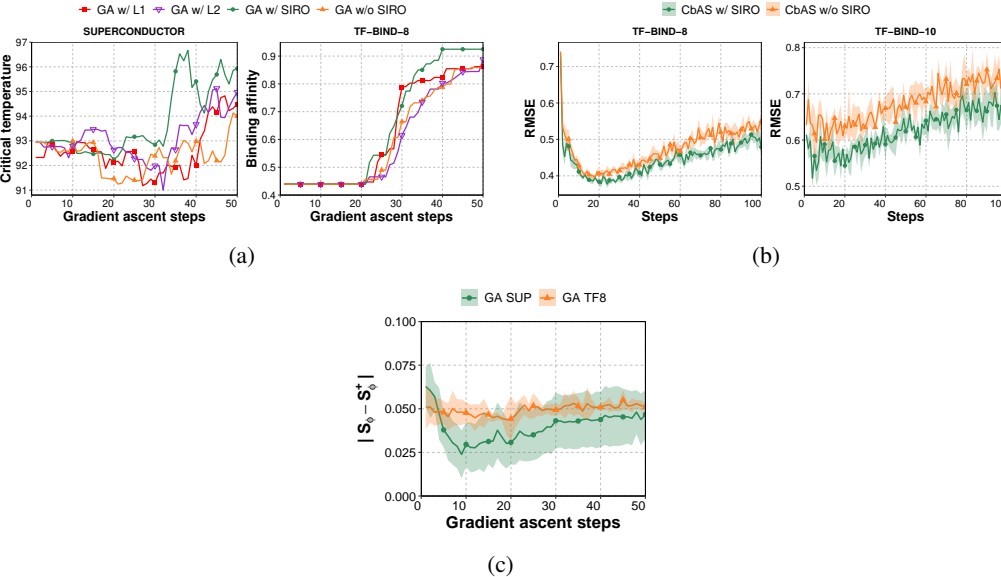

Figure 3: Additional experiments (a) Comparison `SIRO` with $L1$ and $L2$ regularization; Root-mean-square-error (RMSE) between the surrogate model prediction of solution designs and their true objective values with **CbAS** (b); and (c) the tightness of Lemma 2.

### C.2  THE PRECISE OF SURROGATE PREDICTION RESULTS WITH `SIRO`

Our regularization technique is designed to develop a more resilient surrogate model capable of handling minor disturbances and delivering accurate predictions in out-of-distribution regimes. To

assess the precision of predictions using `SIRO` versus the original baseline, we carried out a small-scale experiment. In this experiment, we calculated the root-mean-square-error (RMSE) of surrogate predictions for designs suggested by CbAS (Figure 3b), with the ground truth being the actual objective value determined by the oracle. The outcomes, including the mean and standard deviation of RMSE. The figures showed that the integration of `SIRO` resulted in the surrogate model making more accurate predictions than the original, as particularly indicated by a lower RMSE and reduced standard deviation. This trend was consistent across both tasks, TF-BIND-8, and TF-BIND-10.

## C.3 PERFORMANCE EVALUATION AT 75-TH AND 50-TH PERCENTILE LEVEL

Table 2: Percentage of performance improvement achieved by `SIRO` across all tasks and baselines at the 75-**th percentile** level. **P** denote the achieved normalized performance while **G** denote `SIRO`'s percentage of performance gain over the baseline performance.

| Algorithms | | Continuous Tasks | | | | | | Discrete Tasks | | | | | |
|---|---|---|---|---|---|---|---|---|---|---|---|---|---|
| | | Ant Morphology | | D'Kitty Morphology | | Superconductor | | TF Bind 8 | | TF Bind 10 | | ChEMBL | |
| | | P | G | P | G | P | G | P | G | P | G | P | G |
| $\mathfrak{D}$(**best**) | | 0.565 | | 0.884 | | 0.400 | | 0.439 | | 0.467 | | 0.605 | |
| CbAS | Base | 0.523 ± 0.037 | | 0.797 ± 0.009 | | 0.195 ± 0.014 | | 0.534 ± 0.015 | | 0.496 ± 0.009 | | 0.633 ± 0.000 | |
| | SIRO | 0.522 ± 0.057 | -0.1% | 0.794 ± 0.007 | -0.3% | 0.209 ± 0.012 | +1.4% | 0.531 ± 0.033 | -0.3% | 0.505 ± 0.011 | +0.9% | 0.633 ± 0.000 | +0.0% |
| BO-qEI | Base | 0.607 ± 0.000 | | 0.884 ± 0.000 | | 0.306 ± 0.020 | | 0.439 ± 0.000 | | 0.502 ± 0.007 | | 0.629 ± 0.005 | |
| | SIRO | 0.607 ± 0.000 | +0.0% | 0.884 ± 0.000 | +0.0% | 0.362 ± 0.026 | +5.6% | 0.439 ± 0.000 | +0.0% | 0.502 ± 0.006 | +0.0% | 0.622 ± 0.000 | -0.7% |
| CMA-ES | Base | -0.001 ± 0.014 | | 0.717 ± 0.001 | | 0.389 ± 0.006 | | 0.633 ± 0.015 | | 0.528 ± 0.017 | | 0.633 ± 0.000 | |
| | SIRO | 0.001 ± 0.012 | +0.2% | 0.717 ± 0.002 | +0.0% | 0.391 ± 0.006 | +0.2% | 0.635 ± 0.020 | +0.2% | 0.527 ± 0.013 | -0.1% | 0.633 ± 0.000 | +0.0% |
| GA | Base | 0.180 ± 0.019 | | 0.749 ± 0.026 | | 0.474 ± 0.022 | | 0.802 ± 0.022 | | 0.512 ± 0.007 | | 0.633 ± 0.000 | |
| | SIRO | 0.180 ± 0.016 | +0.0% | 0.755 ± 0.034 | +0.6% | 0.486 ± 0.019 | +1.2% | 0.783 ± 0.039 | -1.9% | 0.508 ± 0.004 | -0.4% | 0.633 ± 0.000 | +0.0% |
| ENS-MIN | Base | 0.220 ± 0.012 | | 0.808 ± 0.013 | | 0.485 ± 0.017 | | 0.821 ± 0.010 | | 0.506 ± 0.009 | | 0.633 ± 0.000 | |
| | SIRO | 0.226 ± 0.013 | +0.6% | 0.819 ± 0.007 | +1.1% | 0.485 ± 0.011 | +0.0% | 0.801 ± 0.008 | +2.0% | 0.507 ± 0.007 | +0.1% | 0.633 ± 0.000 | +0.0% |
| ENS-MEAN | Base | 0.220 ± 0.010 | | 0.828 ± 0.019 | | 0.487 ± 0.026 | | 0.806 ± 0.023 | | 0.503 ± 0.007 | | 0.633 ± 0.000 | |
| | SIRO | 0.220 ± 0.008 | +0.0% | 0.809 ± 0.021 | -1.9% | 0.488 ± 0.009 | +0.1% | 0.795 ± 0.021 | -1.1% | 0.502 ± 0.007 | -0.1% | 0.633 ± 0.000 | +0.0% |
| REINFORCE | Base | 0.169 ± 0.031 | | 0.479 ± 0.187 | | 0.473 ± 0.015 | | 0.564 ± 0.019 | | 0.512 ± 0.009 | | 0.633 ± 0.000 | |
| | SIRO | 0.174 ± 0.032 | +0.5% | 0.712 ± 0.008 | +23.3% | 0.468 ± 0.008 | -0.5% | 0.581 ± 0.023 | +1.7% | 0.511 ± 0.003 | -0.1% | 0.633 ± 0.000 | +0.0% |
| MINs | Base | 0.738 ± 0.024 | | 0.905 ± 0.003 | | 0.363 ± 0.023 | | 0.511 ± 0.015 | | 0.506 ± 0.007 | | 0.633 ± 0.000 | |
| | SIRO | 0.737 ± 0.014 | -0.1% | 0.903 ± 0.003 | -0.2% | 0.372 ± 0.015 | +0.9% | 0.519 ± 0.016 | +0.9% | 0.506 ± 0.006 | +0.0% | 0.633 ± 0.000 | +0.0% |
| COMs | Base | 0.624 ± 0.025 | | 0.884 ± 0.004 | | 0.404 ± 0.018 | | 0.714 ± 0.084 | | 0.512 ± 0.013 | | 0.633 ± 0.000 | |
| | SIRO | 0.614 ± 0.031 | -1.0% | 0.881 ± 0.002 | -0.3% | 0.404 ± 0.010 | +0.0% | 0.746 ± 0.062 | +3.2% | 0.514 ± 0.011 | +0.2% | 0.633 ± 0.000 | +0.0% |

Table 3: Percentage of performance improvement achieved by `SIRO` across all tasks and baselines at the 50-**th percentile** level. **P** denotes the achieved normalized performance while **G** denotes `SIRO`'s percentage of performance gain over the baseline.

| Algorithms | | Continuous Tasks | | | | | | Discrete Tasks | | | | | |
|---|---|---|---|---|---|---|---|---|---|---|---|---|---|
| | | Ant Morphology | | D'Kitty Morphology | | Superconductor | | TF Bind 8 | | TF Bind 10 | | ChEMBL | |
| | | P | G | P | G | P | G | P | G | P | G | P | G |
| $\mathfrak{D}$(**best**) | | 0.565 | | 0.884 | | 0.400 | | 0.439 | | 0.467 | | 0.605 | |
| CbAS | Base | 0.371 ± 0.017 | | 0.737 ± 0.021 | | 0.119 ± 0.017 | | 0.426 ± 0.021 | | 0.456 ± 0.006 | | 0.633 ± 0.000 | |
| | SIRO | 0.381 ± 0.027 | +1.0% | 0.729 ± 0.023 | -0.8% | 0.128 ± 0.006 | +0.9% | 0.420 ± 0.026 | -0.6% | 0.465 ± 0.009 | +0.9% | 0.633 ± 0.000 | +0.0% |
| BO-qEI | Base | 0.568 ± 0.000 | | 0.883 ± 0.000 | | 0.295 ± 0.006 | | 0.439 ± 0.000 | | 0.467 ± 0.000 | | 0.609 ± 0.031 | |
| | SIRO | 0.568 ± 0.000 | +0.0% | 0.883 ± 0.000 | +0.0% | 0.305 ± 0.015 | +1.0% | 0.439 ± 0.000 | +0.0% | 0.467 ± 0.000 | +0.0% | 0.569 ± 0.000 | -4.0% |
| CMA-ES | Base | -0.047 ± 0.005 | | 0.685 ± 0.011 | | 0.378 ± 0.006 | | 0.546 ± 0.011 | | 0.486 ± 0.019 | | 0.633 ± 0.000 | |
| | SIRO | -0.041 ± 0.007 | +0.6% | 0.685 ± 0.010 | +0.0% | 0.379 ± 0.006 | +0.1% | 0.550 ± 0.015 | -0.4% | 0.481 ± 0.012 | -0.5% | 0.633 ± 0.000 | +0.0% |
| GA | Base | 0.140 ± 0.018 | | 0.616 ± 0.124 | | 0.459 ± 0.034 | | 0.613 ± 0.034 | | 0.472 ± 0.004 | | 0.633 ± 0.000 | |
| | SIRO | 0.141 ± 0.020 | +0.1% | 0.583 ± 0.158 | -3.3% | 0.471 ± 0.020 | +1.2% | 0.598 ± 0.043 | -1.5% | 0.468 ± 0.005 | -0.4% | 0.633 ± 0.000 | +0.0% |
| ENS-MIN | Base | 0.183 ± 0.009 | | 0.751 ± 0.018 | | 0.478 ± 0.020 | | 0.659 ± 0.023 | | 0.469 ± 0.003 | | 0.633 ± 0.000 | |
| | SIRO | 0.187 ± 0.011 | +0.4% | 0.758 ± 0.016 | +0.7% | 0.475 ± 0.013 | -0.3% | 0.624 ± 0.028 | -2.5% | 0.469 ± 0.002 | +0.0% | 0.633 ± 0.000 | +0.0% |
| ENS-MEAN | Base | 0.182 ± 0.012 | | 0.777 ± 0.028 | | 0.480 ± 0.030 | | 0.632 ± 0.023 | | 0.469 ± 0.002 | | 0.633 ± 0.000 | |
| | SIRO | 0.182 ± 0.004 | +0.0% | 0.751 ± 0.033 | -2.6% | 0.480 ± 0.009 | +0.0% | 0.629 ± 0.019 | -0.3% | 0.467 ± 0.002 | -0.2% | 0.633 ± 0.000 | +0.0% |
| REINFORCE | Base | 0.137 ± 0.022 | | 0.454 ± 0.192 | | 0.468 ± 0.014 | | 0.440 ± 0.010 | | 0.468 ± 0.007 | | 0.633 ± 0.000 | |
| | SIRO | 0.144 ± 0.029 | +0.7% | 0.697 ± 0.011 | +24.3% | 0.464 ± 0.008 | -0.4% | 0.455 ± 0.022 | +1.5% | 0.472 ± 0.005 | +0.4% | 0.633 ± 0.000 | +0.0% |
| MINs | Base | 0.631 ± 0.033 | | 0.887 ± 0.005 | | 0.333 ± 0.019 | | 0.425 ± 0.010 | | 0.468 ± 0.005 | | 0.633 ± 0.000 | |
| | SIRO | 0.628 ± 0.019 | -0.3% | 0.885 ± 0.002 | -0.2% | 0.346 ± 0.013 | +1.3% | 0.424 ± 0.010 | -0.1% | 0.467 ± 0.006 | -0.1% | 0.633 ± 0.000 | +0.0% |
| COMs | Base | 0.502 ± 0.027 | | 0.862 ± 0.005 | | 0.384 ± 0.020 | | 0.593 ± 0.052 | | 0.474 ± 0.010 | | 0.633 ± 0.000 | |
| | SIRO | 0.489 ± 0.026 | -1.3% | 0.854 ± 0.003 | -0.8% | 0.383 ± 0.013 | -0.1% | 0.634 ± 0.061 | +4.1% | 0.476 ± 0.012 | +0.2% | 0.633 ± 0.000 | +0.0% |

## C.4 TUNING HYPER-PARAMETERS OF THE PROPOSED REGULARIZER

To fine-tune the hyper-parameters of our regularizer, we can use the surrogate of the base method (which was trained on the offline data) as a pseudo-oracle, which helps evaluate design proposals

generated by each specific configuration of the regularizer. For example, we can leverage the surrogate model of **COMs** as a pseudo-oracle to evaluate design proposals generated by the gradient ascent method **(GA)**.

To demonstrate this, we conducted three experiments that mirrored those in our ablation studies in Section 5.3 with the key difference being the use of a pseudo-oracle in place of the true oracle. The results are promising, showing that this technique can successfully discover the same optimal hyperparameters as those determined using the true oracle. Figure 4a shows that the value of sensitivity threshold $\alpha = 0.1$ (GA-0.1) is the best, as previously demonstrated using true oracle in Figure 2b. Likewise, Figure 4b shows that number of perturbation sample $m = 100$ as the corresponding regularized baseline GA-100 is the best among those in that plot. Figure 4c shows the bounds on mean and variance of the perturbation distribution ($[-10^{-3}, 10^{-3}], [10^{-5}, 10^{-2}]$) as the corresponding regularized baseline GA-0 is best. These are the same tuning results found by the oracle in our ablation studies in Figure 2c and Figure 2d. Note that we only use the true oracle in the ablation studies (Section 5.3) which is necessary to show the isolated effect of each of the components. Our other experiments do not use the true oracle for hyper-parameter tuning. It is also noted that the indexing of baseline in Figure 4a, 4b and 4c correspond to different indexing systems of the tuning parameter candidates.

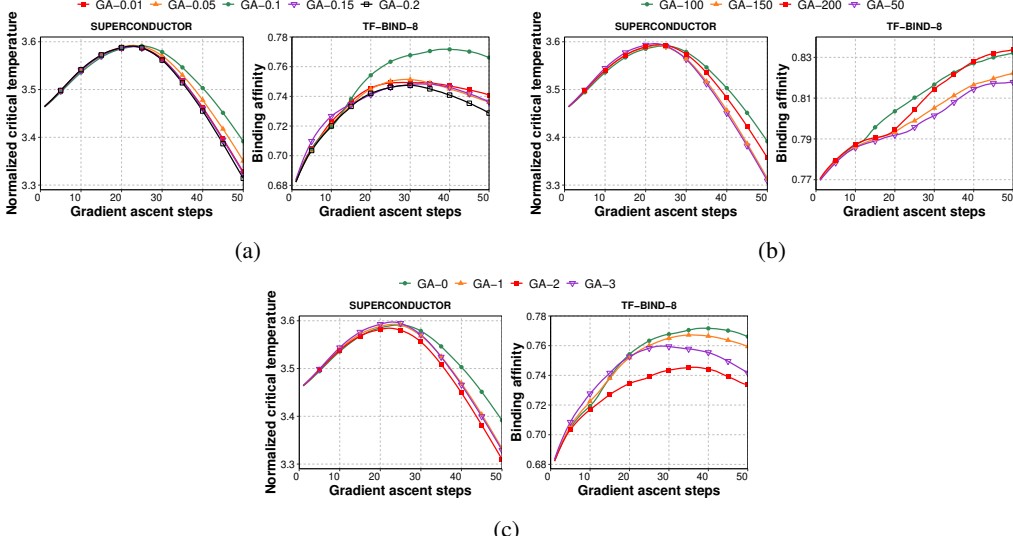

Figure 4: Plots of performance (evaluated by **COMs** pseudo-oracle) variation of **GA** (regularized by SIRO) to changes in (a) values of the sensitivity threshold $\alpha$; (b) changes in the no. of perturbation samples $m$; and (c) changes in value ranges of bound ($[-\omega_{\mu_b}, \omega_{\mu_b}], [\omega_{\sigma_l}, \omega_{\sigma_u}]$) for parameter $\omega$ of the perturbation distribution. These are ($[-10^{-3}, 10^{-3}], [10^{-5}, 10^{-2}]$), ($[-10^{-3}, 10^{-3}], [10^{-6}, 10^{-3}]$), ($[-10^{-2}, 10^{-2}], [10^{-5}, 10^{-2}]$), and ($[-10^{-2}, 10^{-2}], [10^{-6}, 10^{-3}]$) which are indexed with 0, 1, 2, 3 in this figure.

