# OpenReview forum: "SENSITIVITY-INFORMED REGULARIZATION FOR OFFLINE BLACK-BOX OPTIMIZATION"
_ICLR.cc/2024/Conference — ICLR 2024 Conference Withdrawn Submission_

### Official Review · Reviewer_RJQp · 2023-10-25

**Soundness:** 1 poor
**Presentation:** 3 good
**Contribution:** 2 fair
**Rating:** 3
**Confidence:** 3

**Summary:**

This paper tackles the problem of offline black-box optimization. In this problem, a surrogate model is trained to approximate the black-box objective function using a training dataset. The paper proposes a sensitivity measure of the surrogate model. Then, it proposes to train the surrogate model with this sensitivity measure as a regularization term. The goal is to achieve a surrogate model with low sensitivity. Thus, it avoids the problem of out-of-distribution prediction when performing the optimization.

**Strengths:**

The out-of-distribution problem in offline blackbox optimization is important and relevant to the community. The consideration of the model sensitivity seems to be new.

**Weaknesses:**

The proposed approach is not convincing and involves complicated approximations that may not be practical.

1. Could you please explain the rationale behind considering sensitivity across the entire input domain in the definition of the sensitivity measure, while our dataset (and hence our optimization) may only belong to a small manifold of the input domain?

2. The issue raised in the paper is the out-of-distribution prediction of the surrogate model for the black-box function. However, the paper solves it by using another surrogate model without addressing the out-of-distribution prediction of the model. For example, $\Phi(\gamma_i;w)$ is approximated with a neural network. It may also suffer from the out-of-distribution prediction, especially for the high dimensional input $\gamma_i$ which has the same dimension as $\phi$ (the parameters of a neural network). Hence, it raises a concern about the approximation quality.

3. $\Phi(\gamma_i;w)$ is either $0$ or $1$. Is it reasonable to approximate it with a neural network?

4. It is also unclear how tight is the upper bound of the sensitivity in Lemma 2. This issue deteriorates when this upper bound is further approximated using a first-order Taylor expansion. Hence, the resulting impact on the approximation quality is unknown.

5. Even though $\omega$ is trainable, its range still needs to be set manually to avoid the sensitivity measure being vacuous. This may be difficult in practice. Furthermore, other parameters such as $\alpha$ and $\lambda$ also need to be manually set.

**Questions:**

1. The sensitivity measure is defined by considering the expectation over random $x$ drawn from the input domain. What is the distribution of $x$ over the input domain? As $x$ can be a high-dimensional vector, sampling $x$ from its high-dimensional space seems to be very difficult.

2. Why is $\gamma$ drawn from a distribution with non-zero mean $\omega_{\mu}$? Another thing is whether it is reasonable to use the same $\sigma_{\sigma}$ for all parameters, given that some parameters may be very small while other parameters are larger (as these are parameters of a neural network).

3. Please also address the above weaknesses.

---

> ### Author Response · Authors · 2023-11-17
> **Thank you for the detailed review. We have addressed all your questions below.**
>
> **1. The rationale behind considering sensitivity across the entire input domain.** To answer the reviewer’s question, the definition of sensitivity measure is generic but, in our implementation, the expectation over the entire input space is approximated by the empirical expectation of the training input in the offline dataset. Alternatively, we can always find a low-dimensional embedding of the input (e.g., via a variational auto-encoder) and sample on that latent space, whose results can be decoded back to the original space.
>
> **2. Will the approximation quality of $\Phi(\gamma_i; \mathbf{w})$ be affected at out-of-distribution regime.** We would like to assert that the out-of-distribution phenomenon will not happen during the learning of $\Phi$. To see this, note that the erratic out-of-distribution prediction happens on the surrogate of the true oracle because it is trained on non-representative (offline) data observing only values of the oracle at the bottom 40-th percentile -- see COMS -- https://arxiv.org/pdf/2107.06882
>
> On the other hand, **the model $\Phi(\gamma_i,\mathbf{w})$ is used in the context of Eq. (10) which aims to approximate the level set in Eq. (8). This only involves the surrogate model from which we can sample representatively since (unlike the oracle) we have full access to its parameterization. Thus, the out-of-distribution will not happen.** In addition, we note that $\gamma_i$ is a scalar, not a high-dimensional vector.
>
> **3. $\Phi(\gamma_i; \mathbf{w})$ is either $0$ or $1$.** This is a misunderstanding. In our work, the output of $\Phi(\gamma_i; \mathbf{w})$ denote the probability that $\gamma_i$ belongs to the level set in Eq. (8). As such, its value is a continuous value between 0 and 1. It is therefore reasonable to approximate it with a neural network.
>
> **4. How tight is the bound of the sensitivity in Lemma 2?** The tightness of the upper-bound of the sensitivity in Lemma 2 can be seen from the inequality $\mathbb{I}(x \geq \alpha) \leq \min(1, (x/\alpha)^n)$ -- see Eq. (22) & (23) in Appendix B which proves Lemma 2. This inequality is tight because when $x \geq \alpha$, both sides are 1. Otherwise, when $x < \alpha$, $(x/\alpha) < 1$ and so, $(x/\alpha)^n$ is very close to $0 = \mathbb{I}(x \geq \alpha)$ when n is large. Given this, the further use of first-order Taylor expansion should be acceptable as it is a common practice.
>
> To evaluate the tightness of Lemma 2, we undertook a minor experiment to calculate the disparity between $S_\phi$ and $S_{\phi}^{+}$ , specifically $\|S_\phi − S_{\phi}^{+}\|$. The findings of this experiment are visualized in
>
> https://github.com/abcdefghhgfedcba/SIRO/blob/main/rebuttals/tightness/tightness.png
>
> where we present the average and standard deviation across batch data during 50 epochs of training the gradient ascent algorithm (GA) on two tasks: Superconductor and TF-BIND-8. The illustration reveals that the largest divergence $\|S_\phi − S_{\phi}^{+}\|$ is approximately 7.5%, which is sufficiently tight to affirm the accuracy of Lemma 2.
>
> **We will address the remaining questions in the next comment**

---

> ### Author Response · Authors · 2023-11-17
> **Rebuttal (cont.)**
>
> **5. Hyper-parameter Tuning for $\omega$.** Indeed, setting such hyper-parameters is non-trivial. We have however detailed several strategies to do this via building pseudo-oracle based on generating synthetic data, such as those described above in response to Reviewer **kNdc**, which we quote below for your quick review.
>
> To fine-tune the hyper-parameters of our regularizer, we can use the surrogate of the base method (which was trained on the offline data) as a pseudo-oracle, which helps evaluate design proposals generated by each specific configuration of the regularizer. For example, we can leverage the surrogate model of COMs as a pseudo-oracle to evaluate design proposals generated by the gradient ascent method (GA).
>
> To demonstrate this, we conducted three experiments that mirrored those in our ablation studies in Section 5.3 with the key difference being the use of a pseudo-oracle in place of the true oracle. The results are promising, showing that this technique can successfully discover the same optimal hyperparameters as those determined using the true oracle.
>
> This is visualized in the plots below,
>
> **(a)** https://github.com/abcdefghhgfedcba/SIRO/blob/main/rebuttals/hyperparameters_selection/coms_oracle_change_alpha.png
>
> which shows that the value of sensitivity threshold $\alpha = 0.1$ (GA-0.1) is the best, as previously demonstrated using true oracle in Figure 2b. Likewise, the plots at
>
> **(b)** https://github.com/abcdefghhgfedcba/SIRO/blob/main/rebuttals/hyperparameters_selection/coms_oracle_change_n_gamma.png
>
> and
>
> **(c)** https://github.com/abcdefghhgfedcba/SIRO/blob/main/rebuttals/hyperparameters_selection/coms_oracle_change_omega_bound.png
>
> also show that the best hyperparameters are:
>
> No. of perturbation sample $m=100$ as the corresponding regularized baseline GA-100 is best among those in plot (b) and the bounds on mean and variance of the perturbation distribution are $([-10^{-3},10^{-3}],[10^{-5},10^{-2}])$ as the corresponding regularized baseline GA-0 is best among those in plot (c). These are the same tuning results found by the oracle in our ablation studies in Figure 2c and Figure 2d.
>
> Please note that we only use the true oracle in the ablation studies (Section 5.3) which is necessary to show the isolated effect of each of the component. Our other experiments do not use the true oracle for hyper-parameter tuning.
>
> Please also note that the indexing of baseline in plots (a) and (b) correspond to different indexing systems of the tuning parameter candidates.
>
> **6. What is the distribution over random $\mathbf{x}$ drawn from the input domain.** To answer the reviewer’s question, the definition is generic but in our implementation, the expectation over the entire input space is approximated by the empirical expectation of the training input in the offline dataset. Alternatively, we can always find a low-dimensional embedding of the input (e.g., via a variational auto-encoder) and sample on that latent space, whose results can be decoded back to the original space.
>
> **7. Why is $\gamma_i$ drawn from a distribution with non-zero mean?** In our work, we aim to find a model that is most robust against the most adverse (but small) perturbation. Thus, the mean and variance of the perturbation distribution is not set but rather learned adversarially (as described in the paragraph following Eq. (7)). This can also be seen from Algorithm 1, specifically in step 11. As a result of such adversarial training, the mean of the perturbation distribution appears to be non-zero. We describe the overall process in the last paragraph of Section 5.1 and the last paragraph of Section 5.3.
>
> **Overall, we believe we had addressed all your concerns. Please let us know if the response is satisfactory or if you still have follow-up questions for us. We will be happy to discuss further.**

---

> ### Comment · Reviewer_RJQp · 2023-11-18
> **Thank you for the detailed response. However, my concerns persist.**
>
> 1. **the expectation over the entire input space is approximated by the empirical expectation of the training input in the offline dataset.** The paper addresses the challenge of making reliable predictions beyond the scope of offline data. In response to question 2, the authors noted that the offline data is considered **non-representative** of the entire input space. Given this, how do we justify the rationale behind approximating the expectation over the entire input space using the empirical expectation derived from the training input in the offline dataset?
>
> **we can always find a low-dimensional embedding of the input (e.g., via a variational auto-encoder) and sample on that latent space, whose results can be decoded back to the original space.** This might be questionable because of the **non-representativeness** of the offline data. If we can discover a lower-dimensional representation of the input that can be reliably reconstructed into the original space, I am wondering why we do not build a model for the function g using this reduced-dimensional representation and optimize the surrogate with it? It would significantly alleviate the challenge of offline black-box optimization.
>
> 2. I believe the key question here is whether $\gamma_i$ is a scalar or a high-dimensional vector. If it is a scalar, then I agree with the authors that the domain is sufficiently small to be accurately approximated with a neural network. However, the paper defines $\gamma_i = \omega_\mu + \omega_\sigma \epsilon_i$, where $\omega_\mu$ represents the mean of a multivariate Gaussian distribution with the same dimension as the neural network's parameters $\phi$ in Definition 1. Therefore, it would be beneficial to provide additional clarification on why it is a scalar value.
>
> 3. In the first line of Equation 9, the probability within the expectation should be an indicator function of values of either $0$ or $1$ (the only random variable is $\gamma$, so if $\gamma$ is given, $\gamma \in R_\alpha$ is either true or false). Please provide clarification on why it is a probability.
>
> 4. The author asserts that the bound is tight when $n$ is large. However, in the appendix following Equation (23), the author opts for $n=2$ which is not considered a large value.
>
> I'm interested in understanding why we do not incorporate the first-order Taylor approximation from Eq. (14) into the definition of the sensitivity measure in Eq. (4) directly.

---

> > ### Author Response · Authors · 2023-11-19
> > **We believe there are still several misunderstandings of our formulation which leads to the concern on OOD. We will address them below.**
> >
> > Thank you for the quick response. We appreciate that you have given us the opportunity to elaborate further on this important point. We respond to each question below.
> >
> > >**The expectation over the entire input space is approximated by the empirical expectation of the training input in the offline dataset**.The paper addresses the challenge of making reliable predictions beyond the scope of offline data. In response to question 2, the authors noted that the offline data is considered non-representative of the entire input space. Given this, how do we justify the rationale behind approximating the expectation over the entire input space using the empirical expectation derived from the training input in the offline dataset?
> >
> > Thank you for the question.
> >
> > We would like to emphasize that **our method is a performance booster for existing offline optimizers, which already incorporated mechanisms to mitigate the erratic prediction at OOD regime.**
> >
> > However, it is our hypothesis that **the effectiveness of such mechanisms might be reduced if they are defined on a sensitive modeling of the offline data**. Hence, we want to make such modeling less sensitive on the offline data by minimizing the gap between the (average) prediction of the model and its perturbed variant’s, ultimately smoothing out their optimizing behavior. We believe our experiment on a variety of tasks and domains has clearly demonstrated this.
> >
> > >**We can always find a low-dimensional embedding of the input (e.g., via a variational auto-encoder) and sample on that latent space, whose results can be decoded back to the original space.** This might be questionable because of the non-representativeness of the offline data. If we can discover a lower-dimensional representation of the input that can be reliably reconstructed into the original space, I am wondering why we do not build a model for the function g using this reduced-dimensional representation and optimize the surrogate with it? It would significantly alleviate the challenge of offline black-box optimization.
> >
> > We are not suggesting that doing so would enable accurate construction of the entire input space. However, doing so can help generate more synthetic input within the training data regime, helping to enhance the sensitivity conditioning on offline data. Also, this is only a thought, and is not currently a part of our workflow.
> >
> > >I believe the key question here is whether $\gamma_i$ is a scalar or a high-dimensional vector. If it is a scalar, then I agree with the authors that the domain is sufficiently small to be accurately approximated with a neural network. However, the paper defines $\gamma_i = \omega_\mu + \omega_\sigma \epsilon_i$, where $\omega_{\mu}$ represents the mean of a multivariate Gaussian distribution with the same dimension as the neural network's parameters $\phi$ in Definition 1. Therefore, it would be beneficial to provide additional clarification on why it is a scalar value.
> >
> > First, we apologize for the confusion. When we answered your previous questions, we mistook one of your questions as asking about $\kappa_i$ which is the membership label (0/1) corresponding to the perturbation $\gamma_i$ -- defined after Eq. (10) -- which indicates whether the $\gamma_i$ belong to the level set defined in Eq. (8). We also mistyped it into $\gamma_i$, which ultimately causes the misunderstanding.
> >
> > Next, we confirm that $\gamma_i$ is a random Gaussian vector. However, **this does not cause OOD because of two reasons:**
> >
> > **One, we know the true distribution of $\gamma$**: This is parameterized in Eq. (4) and we sampled $\gamma$ (to train $\Phi$) from the same parameterized distribution. Thus, there is no distribution shift between the train and test set on the space of $\gamma$.
> >
> > Two, although $\gamma$ exists in high-dimensional space, its space is factorized into independent 1-D subspaces. To see this, please note that the covariance of the Gaussian distribution over $\gamma$ in Eq. (4) is a scaled identity matrix. As such, with $m = 100$ samples of $\gamma$, we have a set of $m = 100$ scalars per dimension to sufficiently represent its 1-dim Gaussian distribution.
> >
> > OOD only happens when there is a distribution shift or when the samples are to be drawn from a generic high-dimension space with a complex correlation structure. As both of these are not true, we can be sure that OOD does not happen during the training of $\Phi$.
> >
> > **We will address the remaining questions in the next comment**

---

> > > ### Author Response · Authors · 2023-11-19
> > > **Rebuttal (cont.)**
> > >
> > > >In the first line of Equation 9, the probability within the expectation should be an indicator function of values of either 0 or 1 (the only random variable is $\gamma$, so if $\gamma$ is given, $\gamma \in R_\alpha$ is either true or false). Please provide clarification on why it is a probability.
> > >
> > > You are right that the probability within the expectation is indeed an indicator function of value 0/1, i.e. $\mathrm{Pr}(\gamma \in  \mathfrak{R}\_{\alpha}(\phi) | \gamma) = \mathbb{I}(\gamma \in \mathfrak{R}\_\alpha(\phi)) $. Now, computing $\mathbb{I}(\gamma \in \mathfrak{R}\_\alpha(\phi))$ is both costly & non-differentiable. Thus, we instead view this as a logistic regression task where the goal is to predict the label of $\gamma$, which is either zero or one. The idea is we only need to compute $\mathbb{I}(\gamma \in \mathfrak{R}\_\alpha(\phi))$ exactly for a finite sample set of $\gamma$, which can be internalized into a differentiable function $\Phi$ that approximately computes $\mathbb{I}(\gamma \in \mathfrak{R}\_\alpha(\phi))$ for all $\gamma$.
> > >
> > > Our $\Phi$ function is a logistic regressor, which uses a neural net parameterization. We refer to this as the neural re-parameterization -- see the paragraph after Eq. (9). Once learned $\Phi$ tends to assign values closer to 1 for $\gamma$ whose $\mathbb{I}(\gamma \in \mathfrak{R}\_\alpha(\phi)) = 1$ and values closer to 0 for $\gamma$ whose $\mathbb{I}(\gamma \in \mathfrak{R}\_\alpha(\phi)) = 0$. We can consider this a continuous relaxation for the discrete indicator function so that gradient can be propagated through it.
> > >
> > > Our answer to your previous question suggests that $\Phi$ can learn well the relationship between $\gamma$ and its membership label because even though $\gamma$ is in a high-dimensional space, its coordinates are statistically independent which significantly simplifies the learning structure. Otherwise, if $\Phi$ were poorly learned, we would not observe such consistent improvement across a diverse set of tasks and models. Our consistent improvement is shown clearly on Table 1.
> > >
> > > >The author asserts that the bound is tight when $n$ is large. However, in the appendix following Equation (23), the author opts for $n=2$ which is not considered a large value.
> > >
> > > Theoretically, the bound becomes tighter as the value of $n$ increases. However, **we saw that with $n = 2$, the bound is already empirically tight enough, so there is no need to use a larger value of $n$**. On this note, may we know if you have looked at the plot at:
> > >
> > > https://github.com/abcdefghhgfedcba/SIRO/blob/main/rebuttals/tightness/tightness.png ?
> > >
> > > **This plot shows the gap between $S_\phi$ and $S_\phi^{+}$. The latter is an upper-bound of the former. The plot indicates that the gap between the bound and the true value is about 7.5% of the true value, which (we believe) is sufficient to support the claim that the bound is tight even with $n = 2$.**
> > >
> > > Do you agree with us on this instance that a gap of 7.5% is reasonably tight?
> > >
> > > Furthermore, using a larger value for $n$ also does not break our algorithm. We can simply replace it with the new expression in line 13 of Algorithm.
> > >
> > > >I'm interested in understanding why we do not incorporate the first-order Taylor approximation from Eq. (14) into the definition of the sensitivity measure in Eq. (4) directly.
> > >
> > > Unfortunately, we cannot directly apply the first-order Taylor approximation from Eq. (14) to Eq. (4) due to the non-differentiable nature of $S\_\phi$ with respect to $\phi$. Therefore, it is necessary to establish an upper bound for $S_\phi$ that fulfills the differentiability condition. Fortunately, our proposed upper bound, $S_\phi^{+}$, meets this criterion as it is differentiable with respect to $\phi$. Additionally, this bound has been verified to be tight (as in our previous response), making it a suitable alternative for the integration of the approximation.
> > >
> > >
> > > **Once again, thank you for the quick follow-up. We believe we have addressed all your concerns. We are looking forward to hearing back from you.**

---

> ### Comment · Reviewer_RJQp · 2023-11-20
> **Thank you for the response, but they contain many confusions.**
>
> **Sensitivity measure definition**
>
> The response appears to be inconsistent. The initial explanation (and my question) discusses the rationality of approximating the expectation over the input space with that over the offline data. However, in the latest response, the authors argue that the authors hypothesize a sensitive measure should be calculated on the offline data. Additionally, under this hypothesis, the question arises as to why the sensitive measure is not defined as an expectation over the offline data but rather as an expectation over the input space.
>
> **Learning of $\Phi(\gamma_i;w)$**
>
> The authors assert that they misunderstood my question regarding $\kappa_i$ when stating that $\gamma_i$ is a scalar value. This is perplexing, as the initial response from the authors focused entirely on $\gamma_i$, with no mention of $\kappa_i$.
>
> I am also puzzled by the connection between the factorized Gaussian distribution of $\gamma$ and the learning of $\Phi$. In the response, the authors mentioned, "with $m=100$ samples of $\gamma$, we have a set of $m=100$ scalars per dimension to sufficiently represent its $1$-dim Gaussian distribution." Does this imply that if $\gamma$ is $500$-dim, only $500$ samples (**dimension independent?**) of $\gamma$  are needed to find a neural network that approximates $\Phi(\gamma_i;w)$? To approximate $\Phi(\gamma_i;w)$, I believe a substantial number of samples is necessary to adequately cover the distributional support of $\gamma$. This becomes particularly challenging for high-dimensional $\gamma$, even when considering the factorized distribution of $\gamma$. In fact, when the distribution of $\gamma$ is factorized, the support of the distribution becomes even broader compared to the scenario where it is highly correlated.
>
> **$n=2$**
>
> Although there might be additional intricacies in plotting the approximation of $S_\phi^+$, I believe that examining the function $(x/\alpha)^n$ can provide insights into its proximity to zero, particularly when comparing the case of $n = 2$ to a larger value of $n$. Therefore, it appears contradictory that the authors advocate for a large value of $n$ while simultaneously opting for $n = 2$.
>
>
> **First-order Taylor approximation in Eq. (4)**
>
> I do not understand the authors' statement that "we cannot directly apply the first-order Taylor approximation from Eq. (14) to Eq. (4)". I can just utilize the identical Taylor approximation from Equation (14) for Equation (4):
> $|\mathbb{E}[g(x;\phi + \gamma)] - \mathbb{E}[g(x;\phi)]|= |\triangledown h(\phi)^\top \gamma|$
>
> Since, $\gamma$ follows a multivariate Gaussian distribution, $\triangledown h(\phi)^\top \gamma$ follows a Gaussian distribution. Then
>
> $\\text{Pr}\_{\\gamma}(|\\mathbb{E}[g(x;\\phi + \\gamma)] - \\mathbb{E}[g(x;\\phi)]| \\ge \\alpha) = \\text{Pr}\_{\\gamma}(|\\triangledown h(\\phi)^\\top \\gamma| \\ge \\alpha)$
>
> It is only a combination of two univariate Gaussian CDFs. Hence, this greatly simplifies the calculation process.

---

> > ### Author Response · Authors · 2023-11-22
> > **Thank you for the response. We have addressed all your questions below.**
> >
> > >**Sensitivity measure definition**
> >
> > There are no inconsistencies between our original response and the latest response. To be clear:
> >
> > Our 1st response confirms that the definition of sensitivity measure is generic but, in our implementation, the expectation over the entire input space is approximated by the empirical expectation of the training input in the offline dataset.
> >
> > Our 2nd response explains in detail how such empirical approximation of the sensitivity measure will help
> >
> > First, existing methods already incorporate a form of smoothing prediction behavior in the OOD regime.
> >
> > Second, their mechanisms however can still be affected by a sensitive modeling of the offline data.
> >
> > To see this, suppose such mechanisms are brittle at the training data, their brittleness will also be generalized to the unseen test regime. Hence, decreasing the sensitive measure on the modeling of the offline will help even if it is not representative of the input space.
> >
> > Note that we did not claim we prefer the empirical expectation to the exact expectation over the entire input space. The latter is infeasible because the data generation distribution is not known.
> >
> > Furthermore, we want to point out that existing work in model smoothing often comes with a theoretical smoothness measure on the entire input space and then approximate it with an empirical version.
> >
> > For example, please see https://arxiv.org/pdf/2010.01412
> > In particular, 5th line of 1st paragraph in Section 2.
> >
> > >**Learning of $\Phi(\gamma_i;w)$**
> >
> > In our original response, we thought part of the concern is if the output of the neural network were to be a high-dimensional vector, it would become much harder to learn a correct function. Hence, we clarified that $\Phi$ is a probability over a space of two outcomes (0, 1) and as a supporting point & matter of fact, the label $\kappa$ is a 0/1 label.
> >
> > We however made a typo in the last sentence -- in the 2nd point of our original response to the reviewer -- in which we wrote $\gamma$ instead of $\kappa$. Once we correct that single-character typo, all responses are consistent. The reviewer can check this detail regarding $\gamma$ in our released code as well. All is consistent with our explanation here.
> >
> > We apologize again for the confusion.
> >
> > **Next, to the main discussion point, we believe the concern is that $\Phi$ might not be learned properly if a finite set of $m$ samples of $\gamma$ is not sufficient to represent its space.**
> >
> > **Hence, this will be resolved if we can show that $m$ samples of $\gamma$  are sufficient to compute its distribution parameter which succinctly represents its space.**
> >
> > To show this, note that in our Definition 1, $\gamma \sim N(\mu, \sigma^2 * I)$.
> >
> > Thus, suppose we have $m$ samples $(\gamma_1, \gamma_2, …, \gamma_m)$ where each sample is a d-dimensional vector.
> >
> > i.e., $\gamma_i = [\gamma_{i1}, …, \gamma_{id}]$
> >
> > As component of gamma are statistically independent due to the scaled identity matrix, it follows that
> >
> > $log P(\gamma_1, \gamma_2, …, \gamma_m | \mu, \sigma^2)
> > = \sum_i \sum_j log N(\gamma_{ij} | \mu_j, \sigma^2)
> > = \sum_j \sum_i log N(\gamma_{ij} | \mu_j, \sigma^2)$
> >
> > Maximizing the above is equivalent to solving d **independent** MLE task, i.e. $maximize_{\mu_j} \sum_{i=1 .. m} log N(\gamma_{ij} | \mu_j, \sigma^2)$
> >
> > Furthermore, it can also be shown that the variance of the above MLE estimate of the mean for each task, i.e. $Var[MLE(\mu_j)] = \sigma^2 / m = O(1/m)$ with $\sigma^2$ being the true variance.
> >
> > Thus, at $m = 100$, the variance of the estimation will be relatively small, which means the defining information of the space of gamma can be well recovered with $m = 100$.
> >
> > This is possible because the data generation distribution factorizes across individual coordinates of gamma.
> >
> > >**$n=2$**
> >
> > There is no contradiction here.
> >
> > In the original response, we said that the bound becomes tighter as $n$ increases.
> >
> > In the second response, we said that the bound is already sufficiently tight at $n = 2$ so there is no need to increase $n$, and potentially incurs extra computation.
> >
> > **Given the hard evidence that the plot shows a tight bound for $n = 2$, we are not sure why increasing the value of $n$ is important.**
> >
> > **We also do not understand what intricacies are concerning the reviewer. Please let us know.**
> >
> > Otherwise, our rebuttal is consistent with what we wrote in the paper and the released code.
> >
> > **We will address the remaining questions in the next comment**

---

> ### Author Response · Authors · 2023-11-22
> **Rebuttal (cont.)**
>
> >**First-order Taylor approximation in Eq. (4)**
>
> Thank you for this interesting suggestion.
>
> Now, we realize and agree that indeed the Taylor approximation can also be plugged into the above formulation of $S$, making it differentiable.
>
> This is a feasible alternative to our approach and we will inspect it in our revision. However, we do not think this invalidates our approach which is based on an empirically tight bound. It also provides significant improvement across a diverse set of baseline as shown in Table 1.
>
> Overall, it is an interesting exploration for future follow-up but a thorough investigation of this approach is currently out-of-scope for this paper.

---

> > ### Comment · Reviewer_RJQp · 2023-11-23
> >
> > **Sensitivity measure definition** Following the discussion, I have arrived at the conclusion that the sensitive measure's definition should be an empirical expectation from the offline data. It is due to the lack of a clear explanation regarding the approximation of the expectation across the entire input space with the empirical expectation based on offline data.
> >
> > **Learning of $\Phi(\gamma_i;w)$** (the 2nd point in the first response)
> >
> > + **"we thought part of the concern is if the output of the neural network were to be a high-dimensional vector"** (authors' response in quotes): After checking the initial response, there is no reference to the **output being a high-dimensional vector**. Why are the authors introducing the neural network's output at this point?
> >
> > + **"we clarified that $\Phi$ is a probability"** The clarification $\Phi$ is a probability is to address the third point in the first response (which is different from the second point where $\gamma_i$ is claimed to be a scalar).
> >
> > **Learning using $m$ samples of $\gamma$** I do not understand why the authors mention the problem of maximizing $\log P(\gamma_1, \gamma_2, \dots, \gamma_m|\mu, \sigma^2)$. The problem is to learn $w$ in $\Phi(\gamma_i;w)$ using a set of samples $(\kappa_i, \gamma_i)_{i=1}^m$ using Eq. (10)
> >
> > **large values of $n$** I just leave it here for other reviewers and the AC to decide if advocating for larger values of $n$ while setting $n=2$ is a contradiction. Furthermore, employing larger values of $n$ may not result in significantly increased computational burden, contrary to the authors' assertion.
> >
> > **First Taylor approximation** The authors evaded addressing this question in the second response, asserting that directly applying the first-order Taylor approximation from Eq. (14) to Eq. (4) is not feasible. It is only when I explicitly demonstrated this by just directly substituting Eq. (14) into Eq. (4) that the authors admitted its viability. Anyways, this is a significant problem as it implies the possibility of removing many unnecessary approximations and assumptions.
> >
> > In general, there is a lack of sincerity from the authors in addressing the questions, and there are significant problems present in both the paper and the response.

---

> > > ### Comment · Reviewer_RJQp · 2023-11-23
> > > **Clarification on the learning of $\Phi(\gamma;w)$ with 100 samples of $\gamma$**
> > >
> > > I will elaborate my response below in further details.
> > >
> > > > **Learning using $m$ samples of $\gamma$** I do not understand why the authors mention the problem of maximizing $\log P(\gamma_1, \gamma_2, \dots, \gamma_m|\mu, \sigma^2)$. The problem is to learn $w$ in $\Phi(\gamma_i;w)$ using a set of samples $(\kappa_i, \gamma_i)_{i=1}^m$ using Eq. (10)
> > >
> > > Explanation:
> > >
> > > The conversation has revolved around determining the parameters $w$ of a neural network $\Phi(\gamma;w)$ using a set of m training examples $(\kappa_i, \gamma_i)_{i=1}^m$ (following the notations in the paper, where the training input $\gamma_i$ is in the second position, and the label $\kappa_i$ is in the first position). The distribution of $\gamma$ is a multivariate Gaussian distribution with a known mean and a known diagonal covariance matrix.
> > >
> > > To support the claim that we only need around $100$ training examples, in the latest response, the authors discuss the problem of finding $\mu$ that maximizes $\log P(\gamma_1, \gamma_2, \dots, \gamma_m|\mu, \sigma^2)$ (following the authors' argument, so we use $\sigma^2$) for a set of $m$ samples of $\gamma$. By showing that the variance of the estimator of $\mu$ is small with $m=100$ samples, the authors conclude that $m =100$ samples are enough. However, I fail to find any connection between **this problem of of finding $\mu$ that maximizes $\log P(\gamma_1, \gamma_2, \dots, \gamma_m|\mu, \sigma^2)$** and **the problem of finding parameters $w$ of a neural network $\Phi(\gamma;w)$ using a set of m training examples $(\kappa_i, \gamma_i)_{i=1}^m$ where the distribution of $\gamma$ is known**.
> > >
> > > Example:
> > >
> > > Let's assume the validity of the authors' claim that we only require $100$ samples of $\gamma$ (hence, $100$ training examples) to learn $w$ of $\Phi(\gamma;w)$. As the authors' proof remains unaffected by the dimension $d$ of $\gamma$, for the sake of argument, let's set the dimension $d$ to be $200$ (keeping in mind that $\gamma$ shares the same number of dimensions as the parameters of another neural network, hence, $d=200$ is not a large number). It would be surprising if we could find a good estimator for $w$ given that the dimensionality of $w$ should surpass $d=200$ (the dimension of the input to the neural network), yet we only have $m=100$ training examples available.

---

> > > > ### Author Response · Authors · 2023-11-23
> > > > **More Clarification**
> > > >
> > > > Thank you for the quick follow-up. We appreciate it.
> > > >
> > > > From your response, we understand that you have read our proving technique in detail and understood its main argument which motivated the counter example. We appreciate it.
> > > >
> > > > However, comparing $|w|$ with $m$ is not meaningful as the effective amount of information is $O(md)$ instead of $O(m)$ as detailed below. To elaborate, please note that we are free to choose the parameterization of the $\Phi$ function as long as it accepts d-dimensional inputs.
> > > >
> > > > In our implementation, $\Phi$ is a neural network with one hidden layer comprising 2 hidden units, and one output layer, which amounts to a total of $dim(w) = 2d + 5$ parameters.
> > > >
> > > > That is, the neural network $\Phi$ has $O(d)$ parameters and we will have $O(md)$ numbers from $m$ samples of $\gamma$ since each sample of $\gamma$ is d-dimensional. Intuitively, each parameter will have an independent effect on the output due to the statistical independence of the coordinates of the input vector. And as such, with $O(md)$ numbers, we have $m$ numbers to learn each of the $O(d)$ input receptors.
> > > >
> > > > Thus, it is not a surprise that $\Phi$ can be learned well.
> > > > We hope this has addressed your concern.
> > > >
> > > > For better clarity, we will incorporate this explanation into the revised paper. We will detail the parameterization of $\Phi$ explicitly, followed by this discussion.
> > > >
> > > > Thank you for the interesting question.

---

### Official Review · Reviewer_eoHv · 2023-10-30

**Soundness:** 4 excellent
**Presentation:** 3 good
**Contribution:** 3 good
**Rating:** 6
**Confidence:** 3

**Summary:**

This paper proposes to add a sensitivity regularization term to the function surrogate models to improve the performance on offline black-box optimization problem. The authors define the formulation of sensitivity measurement and derive its differentiable version via reparameterization of the Gaussian distribution. Comprehensive experiment on both continuous and discrete benchmarks demonstrate that the proposed regularization term is able to improve the algorithm performance on most cases.

**Strengths:**

1. The idea is clear and the paper is well written.

2. The experiment evaluation and ablation study is extensive and the result is impressive.

**Weaknesses:**

1. According to [1], the Superconductor and ChEMBL benchmarks do not have an exact function oracle, which may inherently have the sensitivity issue, and evaluations on these two benchmarks can not well access the model performance.


[1] Trabucco, Brandon, et al. "Design-bench: Benchmarks for data-driven offline model-based optimization." International Conference on Machine Learning. PMLR, 2022.

**Questions:**

1. The goal of the added regularization term is to make the surrogate function more accurate and less sensitive. In addition to the algorithm performance, can you show some surrogate prediction results with the regularization term?

2. In Table 1 and Ant Morphology task, seems that the variance of the CMA-ES increases with the performance. Can you give some insights of why this happens?

3. In Table 1 and TF Bind 10 task, seems that the REINFORCE with regularization term always find the global optimum. Can you give some insights of this impressive results?

---

> ### Author Response · Authors · 2023-11-17
> **Thank you for the positive review. We have addressed your questions below**
>
> **1. Showing some surrogate prediction results with the regularization term.** Our regularization technique is designed to develop a more resilient surrogate model, capable of handling minor disturbances and delivering accurate predictions in out-of-distribution regimes.
>
> To assess the precision of predictions using SIRO versus the original baseline, we carried out a small-scale experiment. In this experiment, we calculated the root-mean-square-error (RMSE) of surrogate predictions for designs suggested by the gradient ascent method (GA), with the ground truth being the actual objective value determined by the oracle.
>
> The outcomes, including the mean and standard deviation of RMSE, are displayed in
>
> https://github.com/abcdefghhgfedcba/SIRO/blob/main/rebuttals/RMSE/RMSE.png
>
> Throughout 50 steps of gradient ascent, the integration of SIRO resulted in the surrogate model making more accurate predictions than the original, particularly indicated by a lower RMSE and reduced standard deviation. This trend was consistent across both tasks, Superconductor and TF-BIND-8.
>
> **2. Variance of CMA-ES seems to increase with performance.** We do not think this is a consistent pattern. In Table 1, this is only the case with Ant Morphology and D’Kitty. Note that for TF-BIND-10, the variance increases but the performance decreases. Overall, the variance of the regularized optimizer can increase (but not always) because it has more parameters than the non-regularized version.
>
> **3. REINFORCE with regularization always find the global optimum.** Thank you for bringing our attention to this. Upon a closer look into this, we realize that the REINFORCE baseline is somehow sensitive to the randomization of the offline data. We have re-run it for 16 times with more randomization of the offline dataset, the reported performance is $0.964 \pm 0.096$. So, we do not think REINFORCE + SIRO baseline always finds the global optimum. It was a coincidence that REINFORCE + SIRO performs extremely well in one particular dataset generated by Design Bench. When we add more data variation to it, the performance is no longer perfect but it is nonetheless close to the optimal performance.
>
> **We hope the above have addressed all your concerns. Please let us know if you have any follow-up questions for us. We will be happy to discuss further.**

---

> ### Comment · Reviewer_eoHv · 2023-11-20
>
> Thank for your reply about the model prediction and experimental detail.
>
> But I think the result does not convince me of the better prediction of the proposed method. Like I mentioned in the weakness part, the Superconductor benchmark does not have an exact function oracle, which may inherently have the sensitivity issue (seems not addressed in the previous rebuttal).  On the TF-Bind-8 task with an exact oracle, the performance between SIRO and the original model seems at the same level to me.
>
> Therefore, I think the additional result does not help me to better understanding the improved performance of SIRO.

---

> ### Author Response · Authors · 2023-11-23
> **Thank you for the quick response. We have addressed your questions below**
>
> We appreciate the Reviewer for the prompt feedback.
>
> To facilitate a better understanding of SIRO's improved performance, we made an earnest effort to conduct additional experiments on the TF-BIND-8 and TF-BIND-10 tasks. Unfortunately, due to time constraints, we could only complete experiments using the CbAS baseline for TF-BIND-8 and TF-BIND-10. You can access the results at this link: https://github.com/abcdefghhgfedcba/SIRO/blob/main/rebuttals/RMSE/RMSE_CbAS.png
>
> These findings illustrate that the utilization of SIRO aids surrogate models in generating more accurate predictions than the original baseline. We hope that these results adequately address the Reviewer's concerns.
>
> Regarding the issues related to Superconductor task, we agree with the Reviewer's observation that the lack of an exact oracle function could potentially introduce sensitivity issues. In principle, we would prefer to avoid conducting experiments on this task. However, since most prior studies have carried out experiments on this task, we have included the results to provide readers with a comprehensive perspective. We concur with the Reviewer's concern that the Superconductor baseline may not be an ideal choice. However, it's important to note that this concern is rooted in how the benchmark itself was constructed, rather than any shortcomings in our approach.The issues were mentioned before but the benchmark was still published after that. Please refer to the link below for more details:
>
> https://openreview.net/forum?id=cQzf26aA3vM&noteId=_Cs3nvyGs5J .

---

> > ### Comment · Reviewer_eoHv · 2023-11-23
> >
> > Thank you for the additional experiment, and the results are more convincing. I have increased the soundness score.

---

> > > ### Author Response · Authors · 2023-11-23
> > > **Thank you**
> > >
> > > Thank you for upgrading the Soundness score. We appreciate that!

---

### Official Review · Reviewer_bS6c · 2023-11-02

**Soundness:** 4 excellent
**Presentation:** 2 fair
**Contribution:** 3 good
**Rating:** 6
**Confidence:** 3

**Summary:**

This paper investigates offline model-based optimization problems, which build a surrogate model using an offline dataset to approximate an unknown black-box function. This paper proposes an optimizable measurement of model sensitivity and proves its relationship with the brittleness of the model. In order to improve the model’s quality, this paper adds the measurement into the training objective and proposes a bi-level framework to automatically optimize both the measurement parameter $\omega$ and model parameter $\phi$. The key contribution of this paper is the novel measurement of model sensitivity, which can be used to regulate the existing gradient-based surrogate model. Experiment results in this paper appear to show the effectiveness of the improvement brought by model sensitivity regularization.

**Strengths:**

1. This paper is easy to follow and understand.
2. The proposed model sensitivity measurement is well-motivated. The authors provide a novel perspective of measuring the surrogate model’s quality, which theoretically contributes to offline MBO, and it can be scaled to other model-based domains, like BO and offline RL.
3. The procedure of optimizing \omega and \phi is reasonable.

**Weaknesses:**

Experiments are insufficient. Below are the main aspects and my advice:
    1. Lack of tasks. In Design-Bench (Trabucco et al., 2022), there are 8 real-world tasks while this paper only conducts 6 tasks, lacking Hopper-Controller from continuous domain and NAS from discrete domains.
    2. Lack of baselines. This paper only compares SIRO to basic baselines provided by Design-Bench, not containing recent works like Roma (Yu et al., 2021), NoMA (Fu & Levine, 2021), and IOM (Qi et al. 2022), which share a similar framework and sensitivity regularization can be easily applied to. Some works on offline MBO, like COMs (Trabucco et al., 2021), also add regularization terms to the learning objective. If the authors provide experiment comparisons between different regularization methods, the experiments will be more solid and persuading.

**Questions:**

1. Some mistakes in Eq.(6) and Eq.(7) where it should be $\omega = \arg \max_{\omega^{\prime}} \mathcal{S}(\alpha, \omega^{\prime})$ in Eq.(6) and $\lambda\cdot\max_{\omega^{\prime}}S_{\phi^{\prime}}(\alpha, \omega^{\prime})$.
2. In Definition 1, X is defined as the input space. Does “the input space” mean the offline dataset D? It may confuse the readers.

---

> ### Author Response · Authors · 2023-11-17
> **Thank you for the overall positive rating! We have addressed your questions below**
>
> **1. Including more experiments.** Thank you for the suggestions. We report below the **result of regularizing ROMA with SIRO**:
>
> | RoMA | Ant Morphology        | D'Kitty Morphology    | Superconductor        | TF Bind 8             | TF Bind 10            | ChEMBL                |
> |------|-----------------------|-----------------------|-----------------------|-----------------------|-----------------------|-----------------------|
> | Base | 0.553 ± 0.094         | 0.834 ± 0.017         | 0.489 ± 0.022         | 0.665 ± 0.000         | 0.553 ± 0.000         | 0.633 ± 0.000         |
> | SIRO | 0.606 ± 0.059 (+5.3%) | 0.836 ± 0.017 (+0.2%) | 0.498 ± 0.023 (+0.9%) | 0.665 ± 0.000 (+0.0%) | 0.553 ± 0.000 (+0.0%) | 0.633 ± 0.000 (+0.0%) |
>
> As for other experiments regarding NEMO (Fu & Levine, 2021) and IOM (Qi et al., 2022), the authors did not release code so it is impossible to demonstrate the actual impact of SIRO on these optimizers (in their best implementation).
>
> **2. Compare with different regularization methods.** We also provide **extra experiments showing comparison between different regularization methods**:
>
> In particular, we have run additional experiments on the Superconductor and TF-BIND-8 tasks to compare the performance improvement achieved by regularizing the gradient ascent (GA) optimizer with our proposed SIRO regularizer and the traditional L1 and L2 regularizers.
>
> The results are visualized in the below plot hosted on the same anonymized GitHub repository that contains our experimental code:
>
> https://github.com/abcdefghhgfedcba/SIRO/blob/main/rebuttals/L1_L2/L1_L2.png
>
> The results indicate that regularizing GA with L1 and L2 (GA w/ L1 and GA w/ L2) norms is not as effective as regularizing with SIRO (GA w/ SIRO). This is expected since SIRO is specifically designed to condition the output behavior of the model against adversarial perturbation while L1 and L2 only generically penalize models with high complexity, measured by the L1 and L2 norms of their parameters.
>
> **3. Including more baselines.** While we agree with the reviewer that incorporating the suggested baselines are interesting to include and could strengthen our paper, we also believe that not including them in the current manuscript does not pose a weakness to our main claim, as elaborated below:
>
> (1) We do NOT aim to develop a new regularized surrogate that introduces a new offline optimizer competing with existing ones. Instead, we want to develop an optimizer-agnostic regularizer (SIRO) that can be applied synergistically to a diverse set of existing optimizers, boosting their performance as shown in our experiments.
>
> (2) Although the set of existing optimizers used in our experiment is not exhaustive, we also do NOT aim to build a regularizer that works with all optimizers, which might not be feasible. We only claim (in our 2nd contribution bullet) that our developed regularizer works well to boost the performance of a diverse (but not necessarily exhaustive) set of optimizers. Even so, this is still a solid and practical contribution as none of the existing optimizers outperform others in all tasks.
>
> Overall, we agree having those experiments will strengthen the paper (and we have indeed generated new experiment results for some of the suggestions as presented above) but we hope the reviewer will consider our points (1) and (2) above and would not view these missing experiments as weaknesses to our contribution claim.
>
> **4. Typos in Eq. (6) and (7).** Thank you for catching the typo. We have corrected them.
>
> **5. In Definition 1, does the input space mean the offline dataset?**   In Definition 1, $\mathfrak{X}$ is the entire input space that include the example inputs in the offline dataset. We will put a footnote in the paper to make this clear.

---

> ### Comment · Reviewer_bS6c · 2023-11-18
> **Thanks for your response**
>
> Thanks for your response. I still have the following concerns:
>
> 1. Regarding the NEMO code, although it is not open-source, the RoMA project has implemented NEMO. It would be interesting if the author could utilize this implementation available at (https://github.com/sihyun-yu/RoMA/blob/main/design_baselines/safeweight_latent/nets.py) to conduct experiments. Since both codes are based on the design-baselines framework, if the model is applicable, the author can directly make use of it.
>
> 2. In the third point raised by the author, there seems to be a potential misunderstanding of my intention. What I meant to convey was that if the author wishes to highlight the advantages of the regularization term, they can simply incorporate their own regularization term into existing model-based methods. This would effectively demonstrate the significance of their approach. For example, comparing it with COMs by conducting a comparison between Grad. Ascent+SIRO and Grad. Ascent+COMs, in order to investigate the effects of SIRO.
>
> 3. The feedback from Reviewer kNdc regarding the unsatisfactory performance of the 50th percentile and 75th percentile is intriguing. However, the current response lacks sufficient convincing discussion. Further discussion on this issue is requested.

---

> > ### Author Response · Authors · 2023-11-20
> > **Thank you for the quick follow-up. We will address your concern below.**
> >
> > We appreciate that you have given us the opportunity to elaborate further on this important point. We respond to each question below.
> >
> > >Regarding the NEMO code, although it is not open-source, the RoMA project has implemented NEMO. It would be interesting if the author could utilize this implementation available at (https://github.com/sihyun-yu/RoMA/blob/main/design_baselines/safeweight_latent/nets.py) to conduct experiments. Since both codes are based on the design-baselines framework, if the model is applicable, the author can directly make use of it.
> >
> > Thank you for your recommendation. Upon re-examining the RoMA project, we found that while they have made the code for NeMo's surrogate model publicly available, the algorithm itself has not been released. Consequently, we are unable to showcase the specific impact of SIRO on this optimizer (in its best implementation). Therefore, as soon as the code is released, we will incorporate the suggested experiment, as SIRO's compatibility and model-agnostic nature make it well-suited for integration with NeMo. We hope the reviewer will not consider the absence of these experiments as a detriment to the validity of our contributions.
> >
> > >In the third point raised by the author, there seems to be a potential misunderstanding of my intention. What I meant to convey was that if the author wishes to highlight the advantages of the regularization term, they can simply incorporate their own regularization term into existing model-based methods. This would effectively demonstrate the significance of their approach. For example, comparing it with COMs by conducting a comparison between Grad. Ascent+SIRO and Grad. Ascent+COMs, in order to investigate the effects of SIRO.
> >
> > Thank you for the clarification, and we apologize for any confusion earlier. We acknowledge that while COMs is indeed a regularization method, which is specifically tailored for use with the Gradient Ascent (GA). Therefore, the combination of GA and COMs essentially represents COMs alone. Table 1 provides a comprehensive comparison between GA + SIRO and COMs across various tasks:
> >
> > | Task               | COMs Performance | GA + SIRO Performance  | Difference                |
> > |--------------------|------------------|------------------------|---------------------------|
> > | Ant Morphology     | 0.896 ± 0.024    | 0.314 ± 0.034 (-58.2%) | COMs significantly better |
> > | D'Kitty Morphology | 0.937 ± 0.012    | 0.883 ± 0.012 (-5.4%)  | COMs slightly better      |
> > | Superconductor     | 0.483 ± 0.026    | 0.515 ± 0.014 (+3.2%)  | GA + SIRO better          |
> > | TF Bind 8          | 0.946 ± 0.035    | 0.986 ± 0.007 (+4%)    | GA + SIRO better          |
> > | TF Bind 10         | 0.628 ± 0.044    | 0.658 ± 0.072 (+3%)    | GA + SIRO better          |
> > | ChEMBL             | 0.633 ± 0.000    | 0.646 ± 0.002 (+1.3%)  | GA + SIRO better          |
> >
> > COMs outperforms GA + SIRO in only 2 out of 6 tasks, with a significant difference in the Ant Morphology task. To further investigate these findings and due to time constraints, we conducted a quick experiment comparing COMs and SIRO on the Ant Morphology, D'Kitty Morphology and Superconductor tasks, using the REINFORCE baseline. The results are as follows:
> >
> > | Reinforce Task     | COMs Performance | SIRO Performance       | Difference                |
> > |--------------------|------------------|------------------------|---------------------------|
> > | Ant Morphology     | 0.393 ± 0.076    | 0.278 ± 0.014 (-11.5%) | COMs better               |
> > | D'Kitty Morphology | 0.453 ± 0.188    | 0.732 ± 0.003 (+27.9%) | SIRO significantly better |
> > | Superconductor     | 0.300 ± 0.080    | 0.483 ± 0.007 (+18.3%) | SIRO better               |
> >
> > COMs remains superior in the Ant Morphology task by 11.5% but is outperformed by SIRO in D’Kitty Morphology (by 27.9%). From these results, we can conclude that SIRO generally outperforms COMs in most cases and can also be applied on top of COMS to enhance its performance (last row in Table 1 in our paper).
> >
> > **We will address the remaining questions in the next comment.**

---

> > > ### Author Response · Authors · 2023-11-20
> > > **Rebuttal (cont.)**
> > >
> > > >The feedback from Reviewer kNdc regarding the unsatisfactory performance of the 50th percentile and 75th percentile is intriguing. However, the current response lacks sufficient convincing discussion. Further discussion on this issue is requested.
> > >
> > > We would like to explain that our results on 50th and 75th percentile settings are not as good as the result on 100th percentile but we do not think our performance on 50th and 75th percentile are unsatisfactory.
> > >
> > > According to the reported results in Table 2, the no. of cases where there is a slight performance decrease is 16/54. The maximum decrease is 1.9%. That means in 38/54 = 72% of the case the performance is preserved or improved.
> > > The performance strictly increases in 20 cases with a maximum improvement of 23.3%. Overall, the average improvement is 2.245% while the average decrease is only 0.569%
> > >
> > > Likewise, for Table 3, the average improvement is 2.318% while the average decrease is only 0.986%
> > >
> > > We also want to note that the exact wording of Reviewer kNdc is “less impressive” and we think this is not at all strange.
> > >
> > > Following the protocol in the COMS paper, the offline optimizer starts with 128 initial points and performs a number of search steps to arrive at 128 solution candidates. These solutions are sorted in increasing order such that the last point corresponds to the 100th-percentile, the 3rd quarter point corresponds to the 75th-percentile and the middle point corresponds to the 50th-percentile. So, by construction, the result at the 100th-percentile has to be better than the results at the 75th- and 50th-percentile settings. We hope this clarifies the doubt.
> > >
> > > We hope this answers your question. Otherwise, please let us know if you still have follow-up questions.

---

> > > > ### Comment · Reviewer_bS6c · 2023-11-21
> > > >
> > > > Thank you for your response.
> > > >
> > > > I will increase the "Soundness" Score to 4 and lean towards accepting this paper.

---

> > > > > ### Author Response · Authors · 2023-11-21
> > > > > **Thank you**
> > > > >
> > > > > Thank you for upgrading the Soundness score. We appreciate your quick response!

---

### Official Review · Reviewer_kNdc · 2023-11-06

**Soundness:** 4 excellent
**Presentation:** 4 excellent
**Contribution:** 4 excellent
**Rating:** 6
**Confidence:** 3

**Summary:**

This paper presents a framework for offline black-box optimization that accounts for the smoothness/sensitivity to perturbations of the learned surrogate model. The method works by including this sensitivity as a regularization term when training the surrogate model, for which the authors provide a practical scheme and approximation. Many computational benchmarks are used to show that the proposed framework improves existing offline black-box optimization methods.

**Strengths:**

1. The presentation of the regularization methodology and practical implementations is very clear.
2. The results are very impressive and show that the proposed method performs well across a wide variety of problems. Furthermore, the method seems to generalize to improve many algorithms for offline black-box optimization.

**Weaknesses:**

1. While the results show the improvements that come from adding this regularization term, it would be interesting to see how this compares with other methods for regularization, e.g., when tuned properly, standard $L_1$ and $L_2$-norm regularization may help avoid overfitting. More advanced methods may also indirectly penalize non-smoothness.
2. Similarly, the idea of avoiding large output changes subject to a perturbation into the decision variable could be compared against methods assuming the worst-case perturbation, i.e., robust black-box optimization or robust Bayesian optimization [1].
3. The method seems to improve the best-case performance of an algorithm, but performance at the 50th and 75th percentiles is a lot less impressive.

[1] Bertsimas, D., Nohadani, O., & Teo, K. M. (2010). Robust optimization for unconstrained simulation-based problems. Operations research, 58(1), 161-178.

**Questions:**

1. Given that offline black-box optimization assumes you already have a batch of samples, could this information be used to help select the hyperparameters of the proposed regularized? Otherwise, the user is guessing the smoothness of an unknown black-box function.
2. Does regularizing for smoothness make the trained parametric surrogate models behave more similarly to Gaussian processes with standard kernels (i.e., sampling from smooth functions)?

---

> ### Author Response · Authors · 2023-11-17
> **Thank you for the positive rating! We had addressed all your questions below**
>
> **1. Comparison with L1 and L2 regularization.** As requested, we have run additional experiments on the Superconductor and TF-BIND-8 tasks to compare the performance improvement achieved by regularizing the gradient ascent (GA) optimizer with our proposed SIRO regularizer and the traditional L1 and L2 regularizers.
>
> The results are visualized in the below plot hosted on the same anonymized GitHub repository that contains our experimental code:
> https://github.com/abcdefghhgfedcba/SIRO/blob/main/rebuttals/L1_L2/L1_L2.png
>
> The results indicate that **regularizing GA with L1 and L2 (GA w/ L1 and GA w/ L2) norms is not as effective as regularizing with SIRO (GA w/ SIRO).** This is expected since SIRO is specifically designed to condition the output behavior of the model against adversarial perturbation while L1 and L2 only generically penalize models with high complexity, measured by the L1 and L2 norms of their parameters.
>
> **2. Considering similar idea of robust Bayesian optimization in [1].** Thank you for the insightful suggestions. We believe the idea presented in [1] can be repurposed to fit into our context but that will be non-trivial since [1] defines the adverse effect of perturbation in terms of the maximum value a perturbed model can achieve -- see Eq. (4) in [1] -- which does not necessarily measure how susceptible a model is to perturbation.
>
> For example, a low-output model can have its output doubled at the maximum adverse perturbation, but the final output is still less than that of another high-output model whose output only changes by 1% at the worst perturbation. In this case, the proposed algorithm in [1] will favor the former while ours would favor the latter.
>
> This conceptual discrepancy stems from a key difference in focus between our work and [1]. In particular, [1] is set in a context where the regularization is applied directly to the function that needs to be minimized, assuming that it is either known or can be queried.
> As such, it makes sense that the optimizing algorithm in [1] has an implicit bias that low-output models can tolerate more perturbation, which is true in its context. In contrast, our work is set in a more sophisticated context where the regularization is applied to a learning model of the (unknown) function being optimized. As such, unlike [1], we cannot prioritize surrogates with low prediction over surrogates with high prediction.
>
> But nonetheless, we agree that the idea in [1] is very intriguing and we will explore potential adaptation of it into offline optimization in our follow-up work. We note that this is also similar in spirit to the central idea in RoMA [2], another baseline that we had compared against. Thank you again for this interesting suggestion.
>
> [1] Bertsimas, D., Nohadani, O., & Teo, K. M. (2010). Robust optimization for unconstrained simulation-based problems. Operations research, 58(1), 161-178.
>
> [2] Yu, Sihyun, Sungsoo Ahn, Le Song, and Jinwoo Shin. "Roma: Robust model adaptation for offline model-based optimization." Advances in Neural Information Processing Systems 34 (2021): 4619-4631.
>
> **3. Improvement at 50th and 75th percentile.** We agree with the reviewer that results for the 50th and 75th percentile are less impressive than reported results for the 100th percentile. This is potentially because most base methods are also not very effective at those percentiles. It is observed in both Tables 2 and 3 that the 50th and 75th solutions of the base methods often do not improve much over the empirical best. Thank you for pointing this out. This could be one potential venue of focus for our future expansion of the current idea.
>
> **We will address your remaining questions in the next comment.**

---

> ### Author Response · Authors · 2023-11-17
> **Rebuttal (cont.)**
>
> **4. Tuning hyper-parameters of the proposed regularizer.** To fine-tune the hyper-parameters of our regularizer, we can use the surrogate of the base method (which was trained on the offline data) as a pseudo-oracle, which helps evaluate design proposals generated by each specific configuration of the regularizer. For example, we can leverage the surrogate model of COMs as a pseudo-oracle to evaluate design proposals generated by the gradient ascent method (GA).
>
> To demonstrate this, we conducted three experiments that mirrored those in our ablation studies in Section 5.3 with the key difference being the use of a pseudo-oracle in place of the true oracle. The results are promising, showing that this technique can successfully discover the same optimal hyperparameters as those determined using the true oracle.
>
> This is visualized in the plots below,
>
> **(a)** https://github.com/abcdefghhgfedcba/SIRO/blob/main/rebuttals/hyperparameters_selection/coms_oracle_change_alpha.png
>
> which shows that the value of sensitivity threshold $\alpha = 0.1$ (GA-0.1) is the best, as previously demonstrated using true oracle in Figure 2b. Likewise, the plots at
>
> **(b)** https://github.com/abcdefghhgfedcba/SIRO/blob/main/rebuttals/hyperparameters_selection/coms_oracle_change_n_gamma.png
>
> and
>
> **(c)** https://github.com/abcdefghhgfedcba/SIRO/blob/main/rebuttals/hyperparameters_selection/coms_oracle_change_omega_bound.png
>
> also show that the best hyperparameters are:
>
> No. of perturbation sample $m=100$ as the corresponding regularized baseline GA-100 is best among those in plot (b) and the bounds on mean and variance of the perturbation distribution are $([-10^{-3},10^{-3}],[10^{-5},10^{-2}])$ as the corresponding regularized baseline GA-0 is best among those in plot (c). These are the same tuning results found by the oracle in our ablation studies in Figure 2c and Figure 2d.
>
> Please note that we only use the true oracle in the ablation studies (Section 5.3) which is necessary to show the isolated effect of each of the component. Our other experiments do not use the true oracle for hyper-parameter tuning.
>
> Please also note that the indexing of baseline in plots (a) and (b) correspond to different indexing systems of the tuning parameter candidates.
>
> **5. Does regularizing for smoothness make the trained parametric surrogates behave more similarly to Gaussian processes with standard kernels?** Although the Gaussian process (GP) is a prior over a space of smooth functions, its prediction is sensitive to the kernel’s length-scales (one per input component) which are learned based on the offline data, and might not match the actual length-scale in an out-of-distribution regime. This is especially the case in non-static environments where the length-scales tend to vary across the input space. As such, a non-regularized GP will not necessarily behave similarly to a regularized parametric model using our notion of sensitivity in Definition 1 which applies to the entire output space. That being said, we want to emphasize that our proposed regularizer is model-agnostic and can be applied to regularize a GP-based surrogate too. Although it is beside the main point of this paper, it would indeed be interesting to inspect how a regularized GP surrogate performs in an offline optimization context. We will explore this in our follow-up work.
>
> **Thank you again for your suggestion. We really appreciate your insightful comments. We are looking forward to hearing back from you soon. Please also let us know if you have any follow-up questions for us. We will be very happy to discuss further.**

---

> ### Comment · Reviewer_kNdc · 2023-11-22
> **Response to rebuttal**
>
> Thank you for the detailed response. I appreciate the thoughts re. my comments that may have been slightly out of the scope of the original contribution. Though I still have some concerns about the general applicability of the method (e.g., the authors acknowledge performance could be improved at the 50/75 percentiles), I believe the added results improve the contribution of the work. My favorable impression of this paper remains, and I have increased my 'Contribution' score accordingly.

---

> > ### Author Response · Authors · 2023-11-23
> > **Thank you**
> >
> > Thank you for upgrading the Contribution score. We appreciate that!

---

### Author Response · Authors · 2023-11-23
**Discussion summary**

To facilitate the post-rebuttal discussion of our work, we would like to summarize the reviewer-author discussion so far as follows:

**1. Addressing Reviewer Concerns:**

**Reviewer kNdc:**
- We compared our regularizer with L1 and L2 regularization.
- Addressed queries regarding robust Bayesian optimization, improvement at 50th and 75th percentiles, and surrogate behavior resembling Gaussian processes with standard kernels.
- Discussed tuning of hyper-parameters for our proposed regularizer.

**Reviewer bS6c:**
- Responded to concerns about insufficient experimentation by conducting additional experiments for the recent work RoMA.
- Made comparisons with different regularization methods, like L1 and L2, and corrected some typo errors.
- Compared SIRO with COMs regularizer and demonstrated that SIRO typically surpasses COMs in most scenarios.

**Reviewer eoHv:**
- Clarified the variance in CMA-ES and unique REINFORCE performance in the TF-BIND-10 task.
- Addressed specific issues in the Superconductor and ChEMBL tasks.
- Showcased enhanced surrogate prediction results with the inclusion of our regularization term.

**Reviewer RJQp:**
- Resolved misunderstandings about the definition of Sensitivity measure and the learning of $\Phi(\gamma,\omega)$, etc
- Provided insights into hyperparameter tuning for $\omega$
- Presented convincing evidence demonstrating a tight bound for $n=2$, confirming the tightness of our proposed upper bound.

**2. Positive Feedback from Reviewers:**

Reviewer kNdc, Reviewer bS6c, and Reviewer eoHv expressed satisfaction with our responses, agreeing to raise the Contribution and Soundness scores to 4 (Excellent).

---

### Author Response · Authors · 2023-11-23
**Regarding Comments from Reviewer RJQp**

**3. Comments from Reviewer RJQp:**

We appreciate the constructive feedback provided by the reviewer. We have diligently worked to respond to and provide explanations for the issues raised in every comment we've received.

However, we would like to express our disappointment at Reviewer RJQp’s accusation that we lack sincerity.

We are disappointed that the reviewer has accused us of lacking sincerity simply because we did not understand the reviewer’s latest point and even when we had acknowledged it when it was made clearer.

We find such accusation goes against the principle of an academic discussion, which is a process where both sides exchange ideas to seek a common understanding, and to eliminate any misunderstandings (if any) in an amicable manner.

We are also disappointed that while our responses to the reviewer have always been based on facts, the reviewer chose to ignore facts and misinterpret our statement.

For example, we have repeatedly said that **our empirical results show that with $n = 2$ our bound is tight and even though the bound is tighter with larger $n$, we do not see the need to increase it.** However, the reviewer kept interpreting it as a contradiction that we advocate for $n = 2$ over larger $n$. In a recent response, the reviewer even accused us of having intricacies in our experiment plot without any justifications.

In addition, in the last comment, our theoretical analysis which provably shows that the small set of samples of $\gamma$ can represent well its space and generating distribution has also been dismissed. We suspect that the reviewer has not read our response in detail.

Overall, we have been very respectful in our response to the reviewer but at this point, we have to make a clear stand against the accusation that we were being insincere. This is not true.

**We would like to invite all reviewers and AC, as well as all who can see our discussion thread to assess this. All materials of our research work are available for verification.**

---

> ### Comment · Reviewer_RJQp · 2023-11-23
> **Apology to the Authors**
>
> I apologize the authors for the statement about insincerity. This perspective arises from my personal interpretation, and I acknowledge that the authors may have rightfully pointed out that we should not discuss or imply this aspect in academic discussions. I learned a valuable lesson from this experience.
>
> Regarding the experiment plot, my intention was to highlight that analyzing the function $(x/\alpha)^n$ provides a simpler approach. On the other hand, creating a plot for the approximation of $\mathcal{S}_{\phi}^+$ introduces numerous intricacies, including randomness, sample size variations in $\gamma$ and the distribution of $\gamma$. I apologize for any confusion caused by my lack of clarity.
>
> > Although there might be additional intricacies in plotting the approximation of $\mathcal{S}_{\phi}^+$ , I believe that examining the function $(x/\alpha)^n$ can provide insights into its proximity to zero, particularly when comparing the case of $n = 2$ to a larger value of $n$.
>
>
> Nevertheless, my perspective on the paper remains, as I highlighted several major problems in my last response to the authors that have yet to be resolved. This is different from the authors' perspective in the summary.

---

> ### Author Response · Authors · 2023-11-23
> **Further Clarification**
>
> Thank you for the message. We understand and would like to continue our academic discussion.
>
> >**Learning of Phi:**
>
> We appeared to infer that you were concerned about the high-dimensionality issue of learning $\Phi$, which can involve both input and output. Hence, we wanted to make it additionally clear that the output is a scalar. We had this impression because in the original message, your next question to the inquiry on $\Phi$ is about its output. Again, we apologize for the confusion.
>
> >**Learning using $m$ samples of $\gamma$:**
>
> We wrote in our previous message to you that
>
> **Next, to the main discussion point, we believe the concern is that $\Phi$ might not be learned properly if a finite set of $m$ samples of $\gamma$ is not sufficient to represent its space**
>
> **Hence, this will be resolved if we can show that $m$ samples of $\gamma$  are sufficient to compute its distribution parameter which succinctly represents its space**
>
> As we had proved, with $m$ samples of $\gamma$, the MLE model can predict the true mean of $\gamma$ distribution with a small variance. Thus, though the MLE is not the main learning task here, it serves to alleviate the concern that we would need a lot of samples to learn the true space of $\gamma$. Intuitively speaking, if the $m = 100$ samples of $\gamma$ contain enough information about its space and generation distribution, learning $\Phi$ would not be affected by the high-dimensional nature of $\gamma$.
>
> >**Large value of $n$:**
>
> We appreciate your suggestion that increasing the value of $n$ would make the bound tighter. We acknowledged that.
>
> But again, our point is still that with $n = 2$, we achieved good results and the bound looks tight empirically. From that point of view, we do not think it is necessary to tighten the bound further.
>
> In addition, we now understand better what intricacies you were suggesting. But, those intricacies remain with larger values of $n$ so we are skeptical that we can get rid of those by plotting with increasing $n$. Alternatively, if your suggestion is more about a theoretical analysis of $(x/a)^n$ in this context, it is highly non-trivial and is perhaps beyond the current scope. Do you have a suggestion for us on this? We appreciate and highly value your technical suggestions.
>
> Overall, we agree with the reviewer that there is value in investigating the performance of an implementation with higher values of $n$ but this is more about an imperial implementation rather than a main contribution. In addition, we seek the reviewer’s understanding that our compute resource is tight during the rebuttal period so it is hard to repeat the entire set of experiments with larger values for $n$.
>
> >**Taylor Approximation in Eq. (4)**
>
> We have acknowledged that it is a good direction and we will explore it in a future follow-up but for now, it is an alternative solution to the same problem (i.e., using Taylor approximation to reduce the computational cost) and as such, we believe it is orthogonal to our contribution, e.g. even if it does improve the performance further, such improvement does not negate the performance that we have achieved and reported here.
>
> We do agree that your suggestion has the possibility of removing stringent approximations. However, it will require substantial investigation and thus, this idea deserves a separate treatment in another paper.
>
>
> Overall, thank you again for keeping the discussion alive. Your suggestions are potential directions to improve our current paper, which we will investigate in a future follow-up.

---

> > ### Comment · Reviewer_RJQp · 2023-11-23
> > **Thanks for keeping the conversation going**
> >
> > Thank you for continuing the conversation and for accepting my apology.
> >
> > Concerning the learning of $\Phi$, I have provided a brief explanation and an example in my latest response to clarify my concern. Essentially, my hesitation is about the number of unknowns (dimension of $w$) and the number of data points (the samples of $\gamma$). I find it somewhat challenging to accept that having 100 samples of $\gamma$ is adequate, regardless of the dimensionality of $w$, which increases with the input dimension $\gamma$.
> >
> > I acknowledge that my perspective on the values of $n$ may be overly stringent. Therefore, let's assume that $n=2$ is acceptable based on your empirical argument.
> >
> > As for the first-order Taylor approximation approach, I plan to consult with other reviewers and the AC to determine its acceptability. In any case, I hope to see you fully develop the idea that is relevant to your expertise.

---

> > > ### Author Response · Authors · 2023-11-23
> > > **Thank you**
> > >
> > > Thank you for the interesting discussion.
> > >
> > > Despite a few misunderstandings, we believe our discussion has been productive.
> > >
> > > We appreciate your critical questions and valuable insights, especially the intriguing idea that combines the Taylor approximation with the sensitivity measurement.
> > >
> > > In addition, we have also responded to the example you provided. We hope it helps address your concern.
> > >
> > > Please let us know if you have any follow-up questions.
> > >
> > > Thanks,
> > >
> > > Authors